# Risk assessment and perspectives of local transmission of chikungunya and dengue in Italy, a European forerunner

Francesco Menegale [1], Mattia Manica [1], Martina Del Manso[2], Antonino Bella[2], Agnese Zardini[1], Andrea Gobbi [1], Anna Domenica Mignuoli[3], Giovanna Mattei[4], Francesco Vairo[5], Luigi Vezzosi[6], Francesca Russo[7], Federica Ferraro[8], Francesco Maraglino[8], Anna Teresa Palamara [2], Dengue risk assessment working group*, Piero Poletti [1,10] ✉, Patrizio Pezzotti [2,10], Stefano Merler [1,10] & Flavia Riccardo [2,10]

To address the growing frequency, extension, and size of local arboviral outbreaks in Europe we retrospectively analyzed dengue and chikungunya transmission in Italy from 2006 to 2023. We applied generalized additive models to the records of travel-related cases to highlight the spatiotemporal patterns of disease importation, calculated reproduction numbers for six local outbreaks based on autochthonous case data and mapped current transmission risks by applying a computational model that integrates human density, entomological, and climate data. Outbreak locations appear driven by case importation, which is notably higher for dengue – especially from June to October - rather than local transmission risks. Although reporting delays and favorable temperatures allowed onward transmission for several generations from mid-August to mid-November, upon outbreak detection control of transmission was achieved within 15 days. In high-risk areas, significantly longer epidemic risks were found for chikungunya (over 4 months). However, considering observed importation trends, increasingly frequent local dengue outbreaks are expected. Case detection should be prioritized focusing on areas, and in times, where environmental and climate conditions are permissive, regardless of prior outbreaks.

Arboviruses transmitted by *Aedes* mosquitoes are an emerging threat in Europe as highlighted by the increase in geographic extension and cluster size of local transmission events over the past twenty years[1,2]. Italy has experienced, since 2007, the largest epidemics of both dengue and chikungunya in mainland Europe[1,2]. In all those outbreaks,

transmission was sustained by the *Aedes albopictus* mosquito, which is widely established in the country.

The global epidemiology of chikungunya virus (CHIKV) and dengue virus (DENV) presents some similarities such as the geographically widespread transmission (found in most countries with tropical and

[1]Fondazione Bruno Kessler, Trento, Italy. [2]Department of Infectious Diseases, Istituto Superiore di Sanità, Rome, Italy. [3]ASP Cosenza, Calabria Region, Cosenza, Italy. [4]Settore Prevenzione collettiva e Sanità pubblica direzione generale cura della persona, salute e welfare, Emilia-Romagna Region, Bologna, Italy. [5]Regional Service for Surveillance and Control of Infectious Diseases (SERESMI) - Lazio Region, National Institute for Infectious Diseases "Lazzaro Spallanzani" IRCCS, Rome, Italy. [6]General Directorate of Welfare, Regione Lombardia, Milan, Italy. [7]Direzione Prevenzione, Sicurezza Alimentare, Veterinaria - Regione del Veneto, Venice, Italy. [8]Italian Ministry of Health, Rome, Italy. [10]These authors contributed equally: Piero Poletti, Patrizio Pezzotti, Stefano Merler, Flavia Riccardo. *A list of authors and their affiliations appears at the end of the paper. ✉e-mail: poletti@fbk.eu

subtropical climates worldwide[3–5]), and commonalities in symptom presentation. Human infection in both endemic and epidemic settings can present as subclinical or with mild and non-specific symptoms, especially on onset, evolving to be self-limiting in most cases. This makes differential diagnosis on a clinical basis an issue. Due to the lack of widespread routine testing for these and other arboviruses, the number of people affected by these two diseases is likely under-estimated worldwide.

While for CHIKV, recent reviews[6] point towards a decreasing global prevalence over time, DENV incidence has relevantly increased. The World Health Organization (WHO) identified a 10-fold incidence increase globally for DENV between 2000 and 2019, followed by a decrease in 2020–2022 during the COVID-19 pandemic and a sub-sequent upsurge in 2023 and 2024 with several outbreaks occurring in endemic regions and local transmission taking place in areas that were previously unaffected[7–9].

In temperate areas of Europe, small or limited clusters of CHIKV and DENV have been more frequent and larger outbreaks are rare[10]. Low overall case counts lead to limited data availability that in turn limits the strength of the epidemiological evidence required to quantify arboviral epidemic risks in Europe. This highlights the need to integrate the analysis of local outbreaks with epidemiological models leveraging entomological surveillance and climate data to better interpret observed patterns and assess risks at larger spatiotemporal scales. In non-endemic countries, in which the presence of *Ae. albopictus* is widespread, the likelihood of local onward transmission of CHIKV and DENV is linked to three factors: intensity of importation by viremic travelers, favorable ecological conditions for viral transmission including vector abundance, and degree of vector competence of local mosquitoes in transmitting different infections and serotypes.

The first CHIKV outbreak in Italy was detected in 2007[11,12] in the Emilia Romagna region, where over 200 cases of infection were identified over two months in two contiguous villages in the northeast of the country. This event was mostly unexpected at the time and showed that sustained CHIKV transmission was possible in a temperate country. Ten years later, another CHIKV outbreak was observed, with over 400 probable/confirmed cases identified in central Italy, including 80 cases in the capital city of Rome[13,14]. A secondary cluster associated with this outbreak occurred in a village in Southern Italy, causing 100 additional cases[15]. This large outbreak lasted approximately five months.

Local transmission of DENV was first detected in northeastern Italy in 2020, when 11 cases of confirmed dengue serotype 1 (DENV-1) infection were identified in a rural area in the Veneto region[16]. No further local transmission events were reported until 2023, when four unrelated clusters of DENV infection were identified in a period of four months. Affected areas included a small town in northern Italy[17], with 40 laboratory confirmed cases of DENV-1 infection, and multiple cities in Lazio region (central Italy), with sporadic cases of local DENV-2 and DENV-3 transmission and a cluster of 38 confirmed cases of DENV-1 in the city of Rome[18].

To address the limited epidemiological evidence quantifying arboviral epidemic risks in Europe - and given the larger CHIKV and DENV outbreaks observed in Italy compared to other European countries - we conducted a retrospective analysis of *Aedes*-borne transmission in Italy between 2006 and 2023. This includes the analysis of 1577 travel-related and 481 autochthonous cases of CHIKV and DENV identified in the country during this period. We applied generalized additive models to the time series of travel-related cases to highlight spatiotemporal risks of disease importation and to identify temporal changes in notification delays. We estimated and compared the daily reproduction number for CHIKV and DENV, using consolidated time series of symptomatic cases for each outbreak and a model informed by mosquito capture data, incorporating season-specific temperature data to account for climatic factors.

## Results

### Descriptive characteristics of travel related cases and potential outbreak index cases

Between 2006 and 2023, a total of 1577 travel-related laboratory-confirmed cases of DENV and CHIKV infections were reported to the Italian national surveillance system (1435 DENV infections and 142 CHIKV infections). The median age of travel-related cases was similar for the two diseases (36 years, IQR: 27–48; 39 years, IQR: 30–54, for DENV and CHIKV, respectively). Male to female ratios for the two diseases were 1.3 (males: 56.3% for CHIKV vs 55.7% for DENV). (see Supplementary Table S1). The information on the country of exposure was reported for 98.6% of travel related cases, encompassing 87 countries of exposure (see Supplementary Fig. S1). The most fre-quently reported countries of exposure for DENV infections diagnosed in Italy were Thailand, Cuba, India, and Maldives, accounting for 43.0% of imported DENV cases while the most frequently reported countries of exposure for CHIKV infections diagnosed in Italy were India, Dominican Republic, Brazil, and Thailand, accounting for 47.9% of CHIKV imported cases. During the same period, 481 autochthonous laboratory-confirmed cases of CHIKV (93 cases) and DENV (388 cases) were reported. The median age of autochthonous cases was 57 (IQR: 38–69) and 58 (IQR: 42–72) for DENV and CHIKV respectively. Similar male to female ratios were found in the two diseases (males: 52.7% for DENV vs 50.0% for CHIKV). The autochthonous confirmed cases included four sporadic cases of DENV-2 and DENV-3 identified in Lazio in 2023, as well as six major arboviral outbreaks in the country over the past decades: 110 CHIKV cases in Emilia Romagna in 2007, 206 CHIKV cases in Lazio and 72 in Calabria in 2017, 11 DENV-1 cases in Veneto in 2020, and 40 DENV-1 cases in Lombardy and 38 in Lazio in 2023.

Overall, most common symptoms reported by DENV cases were fever (reported by 499 cases), arthralgias (484), skin rash (310), asth-enia (362), and meningoencephalitis (5). Any type of hemorrhagic symptoms including bleeding from gums or nose and petechiae were rare (23 cases, 1.5%). The outcome was favorable for all cases, with no reported fatalities. Most common symptoms reported by CHIKV cases were fever (1424), arthralgias (956), skin rash (708), asthenia (1111), and meningoencephalitis (10). Out of all diagnosed cases, only two fatal-ities were reported, with the remaining patients achieving recovery.

### Temporal trends in case importation and reporting

A strong spatiotemporal heterogeneity characterized the number of travel-related DENV and CHIKV cases identified in Italy between 2006 and 2023 (see Fig. 1). Higher occurrences of imported cases were notified in the Center and North of the country compared to the South, particularly in regions where sustained local autochthonous trans-mission has been reported so far. Despite CHIKV local transmission events generating a higher number of autochthonous cases, data shows a significantly higher number of imported cases of DENV com-pared to CHIKV (1435 vs 142 cases).

Statistical analyses revealed a consistent seasonal pattern in the case importation for both DENV and CHIKV, with a higher inflow of cases identified from July to October for DENV and from June to Sep-tember for CHIKV (see Fig. 1). Significant fluctuations in the yearly number of imported cases were identified for both diseases, with peaks of case importation detected also beyond the recurring seasonal pattern. Despite a reduction during the COVID-19 pandemic, detection of imported DENV infections steadily increased over the 17 years analyzed.

We explored temporal changes in the notification delays asso-ciated with imported cases of DENV and CHIKV, defined as the inter-vals between symptom onset of cases and their reporting to the National Health System. We found that the average notification delays for DENV decreased from 17.1 (95%CI: 14–20.7) days in 2007 to 8.26 (95%CI: 7.1–9.7) in 2023, with an estimated yearly reduction of 4.46% (95%CI: 3.36–5.55%). Although a similar decreasing trend was observed

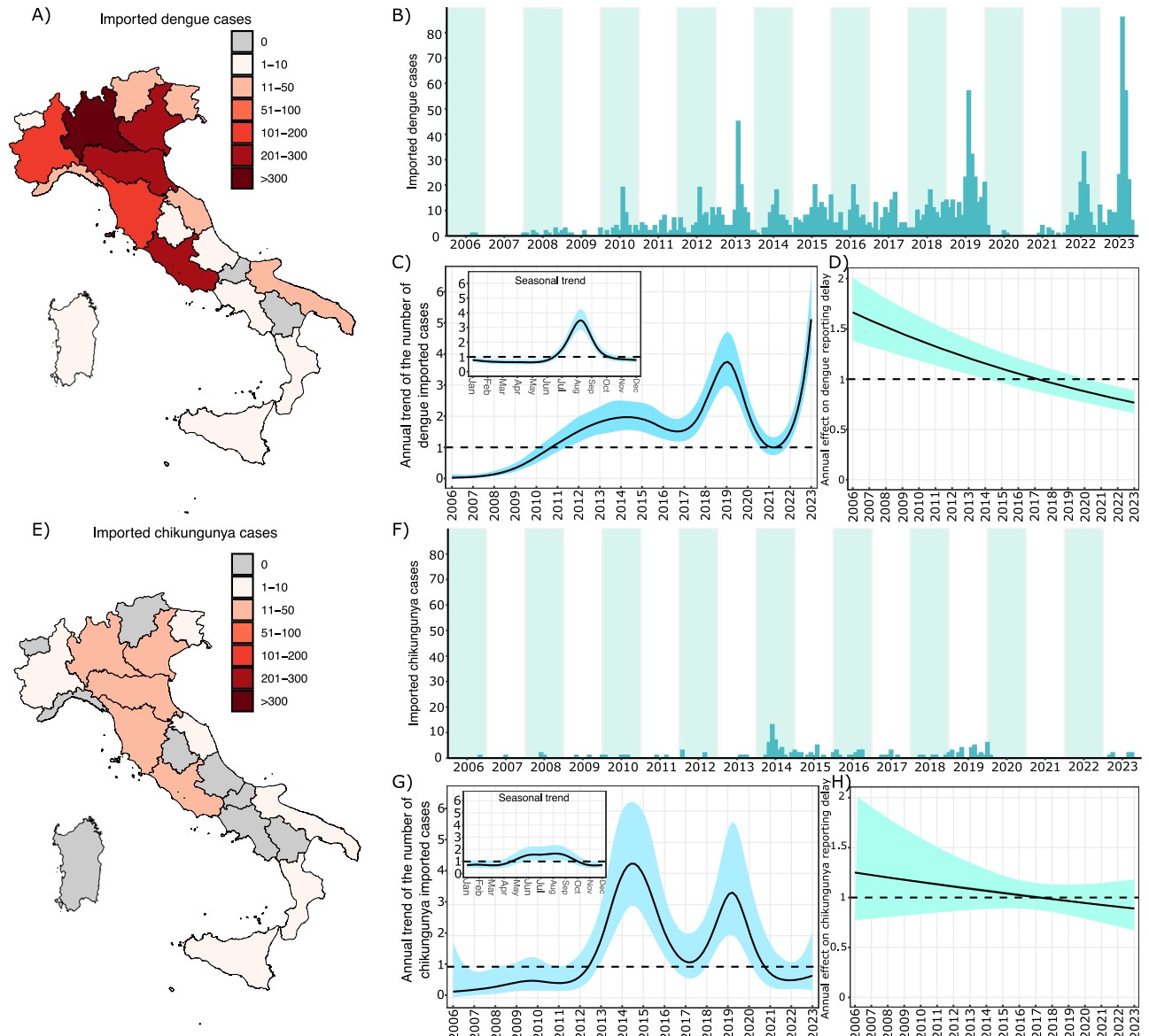

**Fig. 1 | Disease case importation in Italy. A** Map and **B** temporal distribution of travel-related DENV cases from 2006 to 2023. **C** Annual trend in the number of identified travel-related DENV cases, shown as the estimated multiplicative change (solid line) relative to the average number of cases observed over the 2006–2023 period; the shaded area represents the 95% confidence interval (CI) of model estimates. The inset displays the corresponding monthly trend, shown as the estimated multiplicative change (solid line) relative to the average number of cases observed over the year. **D** Annual trend in reporting delays for travel-related DENV cases, shown as the estimated multiplicative change (solid line) relative to the average over the 2006–2023 period; the shaded area represents the 95% confidence interval (CI) of model estimates. **E–H** Same as (**A–D**), but for CHIKV.

for CHIKV cases (average delays: 23.9, 95%CI: 15.8–36.1 days in 2007 and 16.2, 95%CI: 11.3–23.4 in 2023), no significant temporal changes were identified for this infection. For both diseases, notification delays associated with imported cases did not significantly differ from those observed among autochthonous cases with symptom onset occurring after the outbreak detection. In contrast, we found 2.08 (95%CI: 1.63–2.77) times longer delays for autochthonous cases with symptom onset preceding the outbreak identification (see Supplementary Information).

### Transmissibility from surveillance of human cases

We estimated the daily net reproduction number ($R_t$) for CHIKV and DENV, leveraging the time series of symptomatic cases retrospectively consolidated for each autochthonous outbreak (see Fig. 2). The peak of $R_t$ was identified between mid-August and the end of September, ranging on average from 0.96 to 2.06 for DENV and from 2.25 to 4.14

for CHIKV. We found that $R_t$ declined shortly after the outbreak detection and the consequent implementation of vector control measures, dropping below the epidemic threshold ($R_t = 1$) within 15 days. The DENV-1 outbreak that occurred in Lazio during 2023 stands out as an exception to this trend, with onward transmission ($R_t$–1) persisting until mid-November. More generally, a prolonged risk of onward transmission ($R_t > 1$) was identified in autumn for outbreaks that occurred in the Center and South of Italy. For the CHIKV outbreak that occurred in Emilia Romagna in 2007, a decline of $R_t$ was also observed shortly before the outbreak identification.

### Risk of onward transmission and climatic trends

We mapped the spatiotemporal changes in the risk of onward transmission of DENV and CHIKV at the national level, using a model informed by entomological surveillance data. Resulting estimates of the daily reproduction number $R_0$ at a spatial resolution of

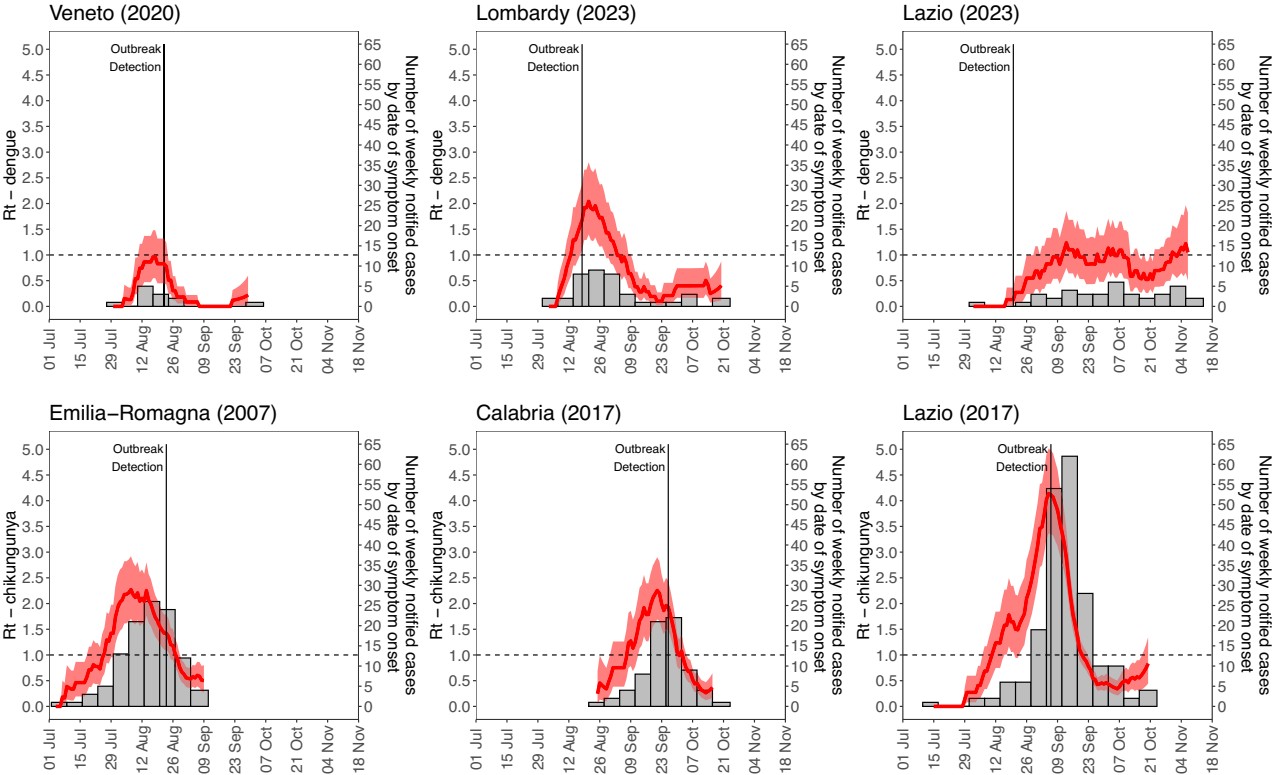

**Fig. 2 | Transmission from surveillance of human cases.** Mean (solid line) and 95% CI (shaded area) estimates of the net reproduction number $R_t$, based on the time series of notified cases by date of symptom onset across the six outbreaks of DENV (top) and CHIKV (bottom) occurred in Italy between 2006 and 2023. Gray bars represent the weekly number of notified cases by date of symptom onset.

100 m × 100 m clearly show that areas affected by past CHIKV and DENV outbreaks were associated with a significant risk of onward transmission in August (see Fig. 3). However, risk maps also suggest that many other areas across the country were similarly at risk of experiencing sustained autochthonous transmission during the summer and in early autumn.

For each arboviral outbreak occurring in Italy between 2007 and 2023, we aggregated $R_0$ estimates associated with populated patches falling within a square area of 900 m × 900 m centered around the likely site of exposure of autochthonous cases. The resulting $R_0$ values are in good agreement with peaks of $R_t$ estimated from surveillance of human cases (see Fig. 4). In general, our estimates show that the dates of symptom onset of the first autochthonous case identified for each outbreak occurred during a period favorable for onward transmission (i.e., $R_0 > 1$ for at least one generation time[8,14]). An exception to this pattern was found in Rome, where - in line with $R_t$ estimates - the average $R_0$ of DENV remained slightly below the epidemic threshold. However, the marked spatial heterogeneity characterizing $R_0$ estimates in this highly urbanized area is consistent with the scattered distribution of cases identified during the outbreak (see Supplementary Fig. S7). Concerning the 2007 CHIKV outbreak in Emilia Romagna, we found a decline of $R_0$ at the end of July 2007 as resulting from changes in temperature conditions. This may partially explain the corresponding decline of $R_t$ found before outbreak detection (see Fig. 2).

The analysis of the estimated $R_0$ in August over the years suggests strong inter-annual variations in the risk of onward transmission for both infections, with a higher potential of transmission in 2017 and 2023 compared to the other outbreak years (see Fig. 5). This pattern was found to be more pronounced in the northwest and central regions of Italy, where the number of detected imported cases was observed to be higher. Although a progressive increase in the annual number of imported cases was observed (see Fig. 1), a consistent increase in the risk of onward transmission after case importation was not observed over the last 16 years (see Fig. 5). Overall, a significantly higher risk of onward transmission was consistently found across the study period for CHIKV compared to DENV (see Fig. 6).

Coastal villages and peripheral areas close to highly urbanized cities were identified as having a higher risk of onward transmission. In these areas, the duration of epidemic risk for CHIKV was estimated to last more than 10–12 generation times (corresponding to 20–24 weeks; CHIKV generation time ~14 days[14]). A shorter duration of epidemic risk was estimated for DENV, with most areas at risk of onward transmission for less than 2 generation times (~5 weeks; DENV generation time ~18 days[8]).

## Discussion

This paper presents a comprehensive analysis of the transmission risk of CHIKV and DENV in Italy, considering available epidemiological records as well as climate, ecological, entomological, and human density elements. Italy is among forerunner countries in Europe for CHIKV and DENV transmission[1,2,19,20], and among the countries where climate change appears to be more evident[19]. It is also the country with a longer history of extensive colonization by *Ae. albopictus*[21], possibly anticipating scenarios that could be observed in other European countries currently experiencing the introduction or increasing colonization of this mosquito species[22].

Our study showed time and place heterogeneity of imported DENV and CHIKV incidence in Italy, with higher numbers being reported in northern and central parts of the country and a strong seasonality leading to higher incidence during the warmer months of the year. Although different reporting rates across regions could not be excluded, the geographical element of diversity might be explained by socio-economic factors as more urbanized, better served (e.g., by

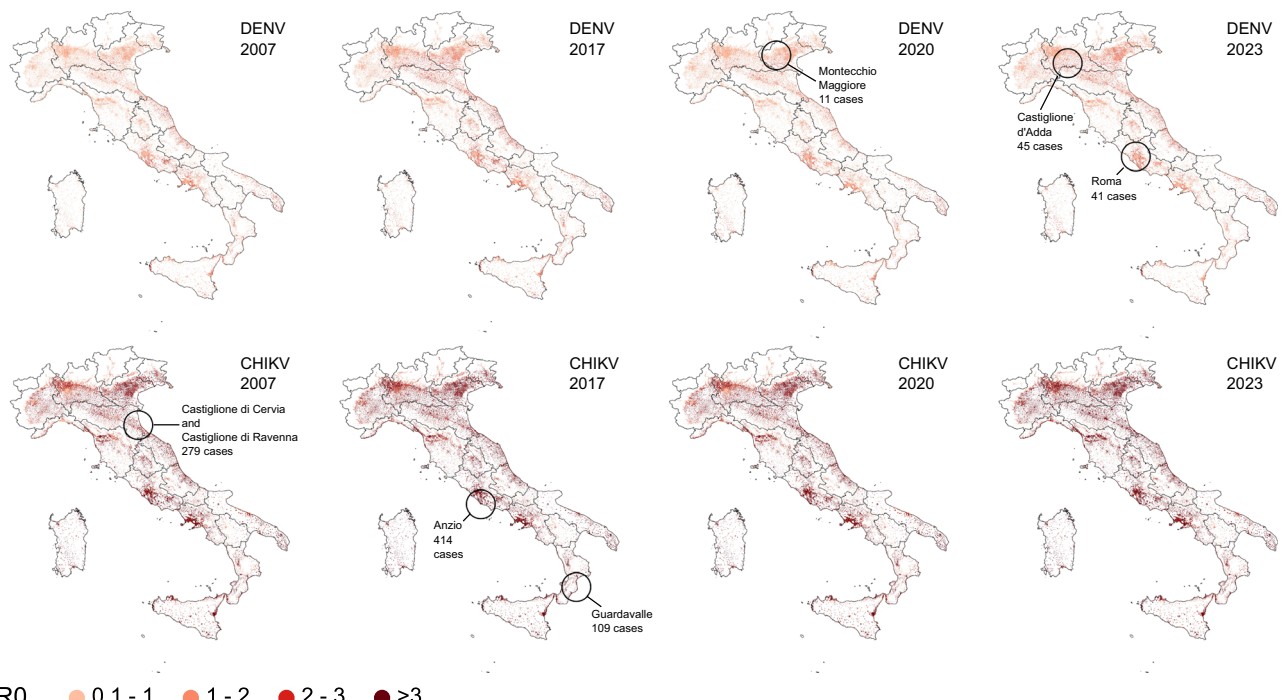

R0   0.1 - 1    1 - 2    2 - 3    >3

**Fig. 3 | Risk of onward transmission from entomological data.** Estimated mean basic reproduction number ($R_O$) for DENV (top) and CHIKV (bottom) in Italy at the beginning of August during the outbreak years, assuming *Ae. albopictus* as the primary vector species. Estimates are provided at a 100 m × 100 m spatial resolution and are shown only for areas with a population density of at least ten inhabitants per hectare and where the estimated $R_O$ exceeds 0.1. Values are displayed as proportional to the population size to highlight potential epidemic risks and represent the average estimate of 500 model runs. Maps also show the locations and sizes of historically observed outbreaks.

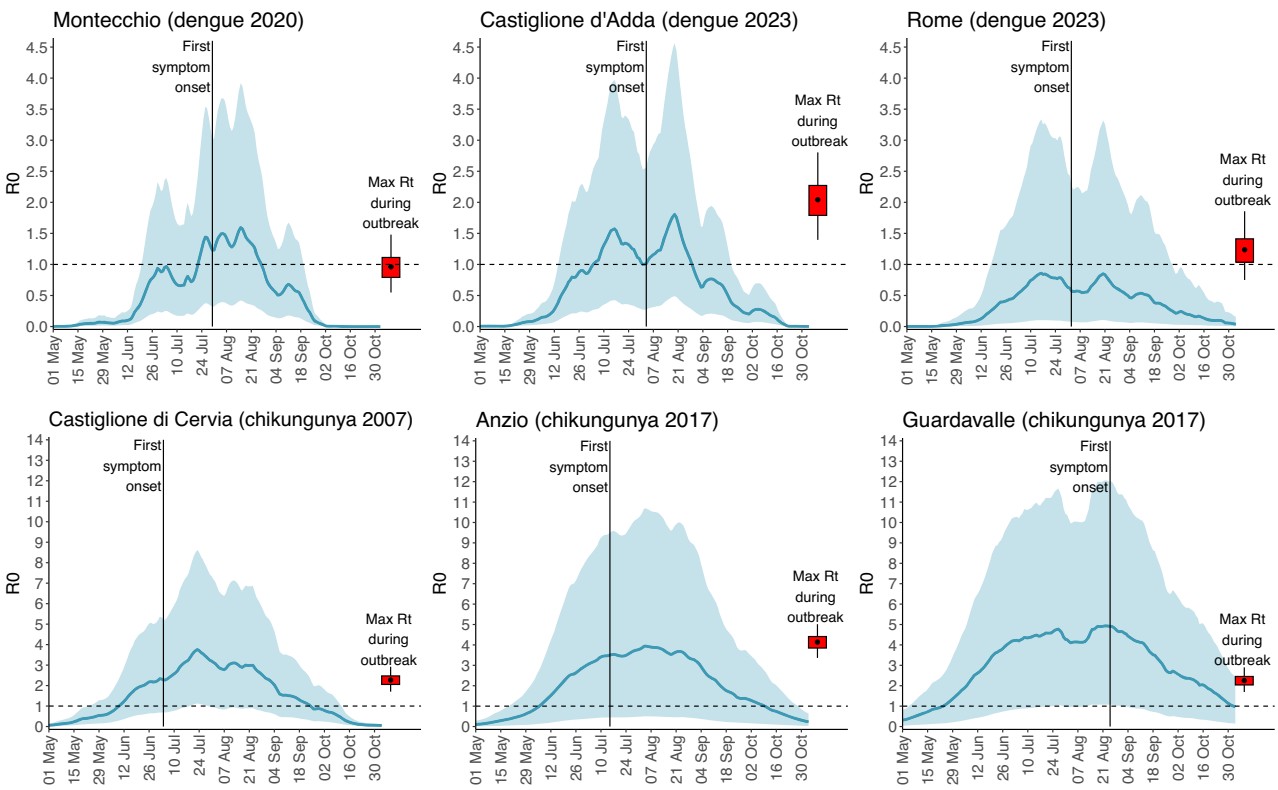

**Fig. 4 | Comparison of transmission estimates from human and entomological data.** Mean (solid line) and 95%CI (shaded area) estimates of DENV and CHIKV $R_O$ obtained by applying the risk model for onward transmission across various outbreak locations during the outbreak years. The red boxplot represents the estimated posterior distribution (2.5th, 25th, 50th, 75th, and 97.5th percentiles) of the peak $R_t$ value estimated from the corresponding time series of human confirmed cases by date of symptom onset.

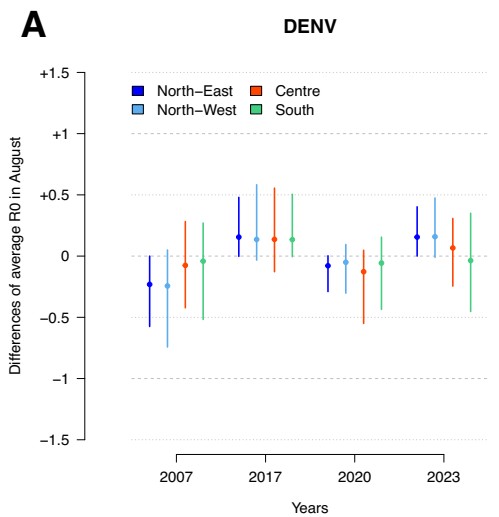

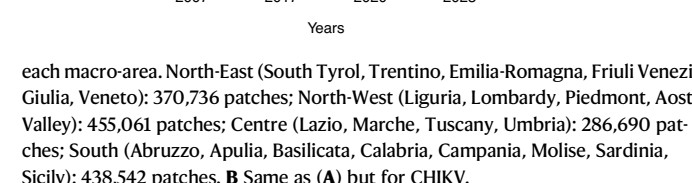

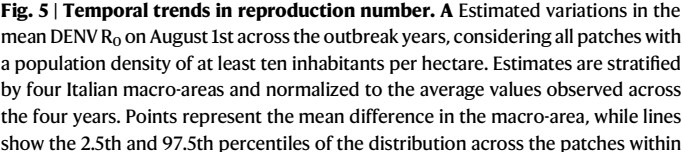

**Fig. 5 | Temporal trends in reproduction number. A** Estimated variations in the mean DENV $R_0$ on August 1st across the outbreak years, considering all patches with a population density of at least ten inhabitants per hectare. Estimates are stratified by four Italian macro-areas and normalized to the average values observed across the four years. Points represent the mean difference in the macro-area, while lines show the 2.5th and 97.5th percentiles of the distribution across the patches within each macro-area. North-East (South Tyrol, Trentino, Emilia-Romagna, Friuli Venezia Giulia, Veneto): 370,736 patches; North-West (Liguria, Lombardy, Piedmont, Aosta Valley): 455,061 patches; Centre (Lazio, Marche, Tuscany, Umbria): 286,690 patches; South (Abruzzo, Apulia, Basilicata, Calabria, Campania, Molise, Sardinia, Sicily): 438,542 patches. **B** Same as (**A**) but for CHIKV.

international airports) and economically stronger areas have an overall North-South gradient in Italy. The seasonal diversity we observed for both DENV and CHIKV is likely the consequence of prevailing travel patterns during the summer holidays in concurrence of large global epidemics[23]. Consistently with international trends, in the study period (2006–2023), we observed a significant general uptrend in the number of imported DENV cases in Italy and a much higher incidence of DENV importation compared with CHIKV importation. Reported countries of exposure to DENV and CHIKV infections were in line with those documented by the European Centre for Disease Prevention and Control (ECDC) for travel-related CHIKV and DENV cases ascertained in the European Union from 2019 to 2023[24,25].

The observed pattern of local transmission events of DENV and CHIKV in Italy is consistent both with the epidemiology of imported infections and with local transmission potential, peaking from mid-August to the end of September. More prolonged epidemic risks were identified in the Center-South of the country, where locally sustained transmission can be protracted until mid-November due to persisting favorable temperature conditions. As observed in previous studies[26,27], our study provides additional evidence that the risk of onward transmission is significantly higher and longer for CHIKV compared with DENV.

Coastal villages and peripheral/sub-urban areas close to highly urbanized cities were identified as having a higher risk of arboviral transmission, due to higher vector-to-host ratio[28] and plasticity of *Ae. albopictus*[26,28,29]. All locations where local transmission of CHIKV and DENV has been observed in Italy were identified as high-risk areas in our analysis. However, also many other areas were found with similar ecological suitability and therefore could be potentially at risk. This suggests that the exact location of historical outbreaks may have been determined by the incidental importation of cases rather than by a localized higher risk of onward transmission. This implies orienting public health prevention and detection activities on areas with similar permissive environments regardless of whether outbreaks have occurred there before.

Overall, we found an improved timeliness in the detection of DENV cases that might be explained by an increasing awareness of arboviral infections. Nevertheless, existing delays in notification and response are still sufficient to allow onward transmission for several generation times.

Once detected, the protocol for outbreak management is standardized and a package of measures and control interventions implemented regardless of the size of urban settlements or level of urbanization. However, a higher difficulty in controlling transmission was observed in large urban settings (e.g., Rome) where the risk is more heterogeneous than in smaller cities, a higher frequency of importation is expected within the same season, and multiple transmission events have been observed to occur simultaneously. Key factors that may limit the effectiveness of control interventions include difficulties in accessing private properties and the resistance of *Ae. albopictus* to pyrethroid insecticides[30–32]. Additionally, in highly populated cities, human mobility complicates identifying exposure sites and areas of sustained transmission, making it harder to identify at-risk populations and implement effective vector monitoring and control activities. This notwithstanding, our results show that control of local outbreaks, defined as $R_t$ below 1, was mostly reached within one generation time following the outbreak detection (index case). This finding underscores the importance of early identification of autochthonous cases, as initial detection often occurs weeks after introduction, with a suspected primary case of local transmission identified in only 3 out of 6 outbreaks analyzed[10,12,16,28]. A recent review highlights the challenge of detecting primary imported DENV/CHIKV cases in Europe, with median delays of over 35 days between the symptom onset of the primary case and the diagnosis of the index case[10].

In Italy, detection is predominantly limited to symptomatic cases. Serological surveys after the 2007 Italian CHIKV outbreak estimated a symptomatic ratio of 82% or more[33,34], but similar data for DENV in non-endemic settings is lacking. Available data from endemic regions suggest a symptomatic ratio of around 20%, while voluntary screenings during the 2023 DENV outbreak in Lombardy found a ratio of 80%[8]. Further studies are needed to determine whether this observation results from under-detection or if the clinical presentation differs between populations rarely exposed to DENV and those with a long history of endemic exposure. In temperate regions, lack of awareness of physicians on *Aedes*-borne arboviral diseases may contribute to an under-diagnosis and in delaying the detection of local outbreaks. The description of clinical symptoms reported here, combined with the key seasonal timing and travel history of imported cases reported in

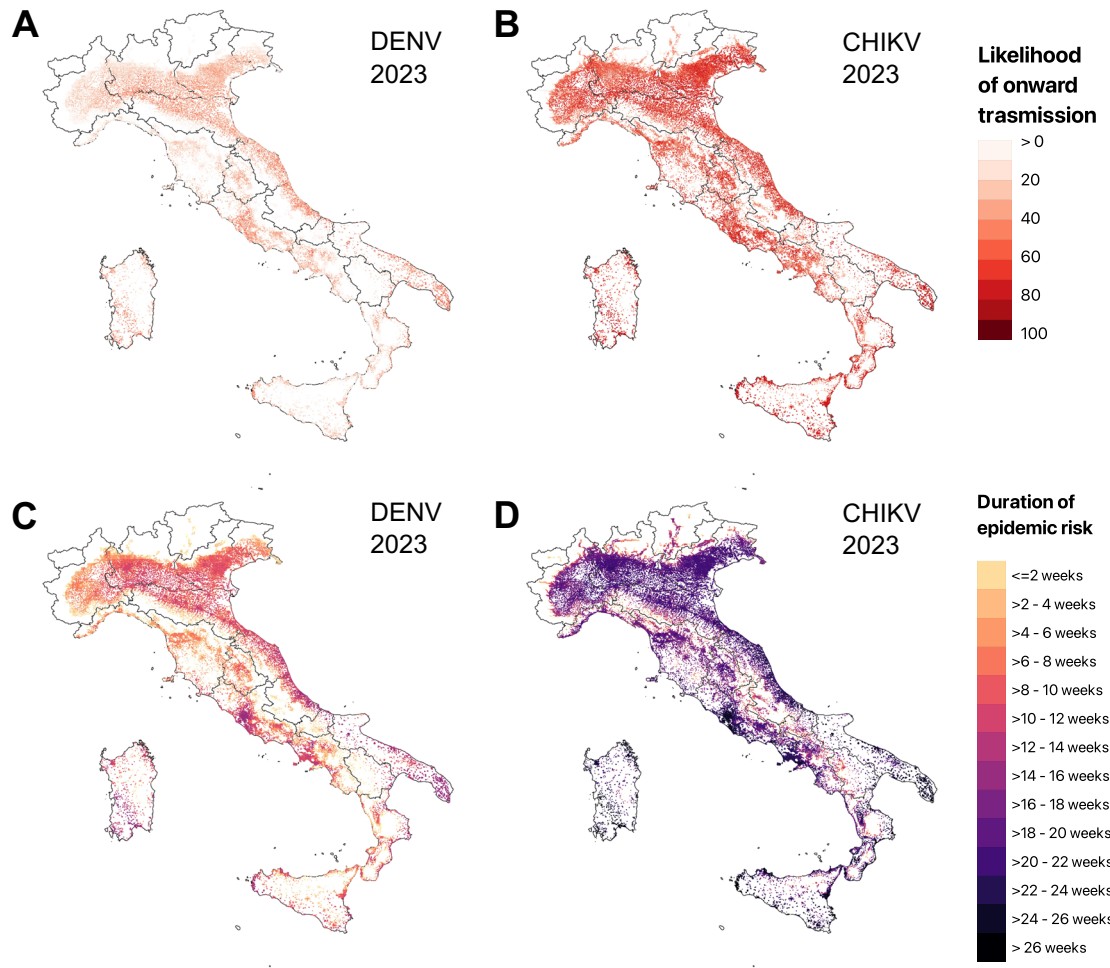

**Fig. 6 | Likelihood of onward transmission and duration of epidemic risk in Italy. A** Estimated likelihood of onward DENV transmission in Italy on August 1st, 2023, shown only for areas at risk of autochthonous transmission ($R_0 \geq 1$). Patches are displayed as proportional to population size to highlight potential epidemic risks and represent the average estimate of 500 model runs. **B** Same as (**A**) but for CHIKV. **C** Estimated duration of epidemic risk for DENV, defined as the number of consecutive days during which the estimated $R_0$ exceeds the epidemic threshold of one. Estimates are shown only for areas at risk of autochthonous transmission for at least one day and represent the average estimate of 500 model runs. **D** Same as (**C**) but for CHIKV.

the past, may increase awareness and enhance the probability of recognizing symptoms of arboviral infections, particularly during the season of high vector activity. Given the challenges in case identification[10], and recent advancements in the detection of arboviruses in wastewater samples[35,36], wastewater surveillance might become a complementary tool to overcome the limitations posed by asymptomatic and other non-recognized infections. Airplane wastewater monitoring could help identify periods of highest importation risk[37], while community-level wastewater surveillance may assist in detecting potential viral spread[35], enhancing early warning capabilities triggering epidemiological investigations and containment efforts.

Even if the risk of onward transmission ($R_0 > 1$) for CHIKV and DENV does not show a clear yearly trend since 2007, we can expect an increase in the frequency of local DENV transmission events due to clear increases in the importation trends. Based on concurrent evidence from human and entomological surveillance, we expect that local DENV transmission events will be more frequent than CHIKV local transmission events. However, once initiated, we expect onward transmission of CHIKV to be more effective and determine larger outbreaks due to the combination of a higher $R_0$ and a shorter generation time.

Although season-specific effects remained crucial in the analyzed outbreaks, we presented evidence that CHIKV and DENV transmission in urban areas of Italy is possible and that transmission risks in

southern regions may extend into late autumn. We also showed that, in Italy's long, warm autumns, outbreaks can persist for months even when Rt drops below the epidemic threshold, with transmission typically ending as temperatures decrease. This has resulted in relatively large CHIKV and DENV outbreaks compared to other EU/EEA countries. As suggested by 2024 data currently under consolidation[20,38], epidemic risks are increasing, with several concomitant local clusters reported, including one with over 100 locally acquired DENV infections[9]. As the geographic expansion of *Ae. albopictus* is expected across large portions of Europe[26], further efforts are needed to quantify how changes in climate, urbanization, vector capacity, and inter-country mobility may influence future arboviral transmission risks.

This paper presents the following limitations. Estimates of $R_t$ were obtained under the implicit assumption of a constant ascertainment ratio within a season, and notification delays were computed as the time between the symptom onset of imported cases and their reporting to the National Surveillance System. The latter quantity may represent an overestimation of the time required for case detection by local health authorities, who are responsible for implementing reactive interventions upon identifying a suspected or confirmed case. The lack of a significant decrease in the notification delay for CHIKV might be influenced by the small sample size of chikungunya cases in our data (142 cases), possibly driven by a lower awareness of chikungunya risks compared with dengue among general practitioners. However, the

level of underestimation in number of infections might be different between these two pathogens. More generally, notified travel-related cases represent only a fraction of total imported cases due to under-reporting, which may be strongly influenced by cocirculation of diseases characterized by similar symptoms. Biases arising from spatial and temporal heterogeneity in case ascertainment ratios, including differences in screening practices, cannot be excluded and may affect the interpretation of the data. Mapping of disease risks at the national level was performed by assuming the resident population's size as a proxy for the number of hosts exposed to viral infection. This might have introduced biases in estimating the risk of onward transmission at a micro-scale level, as individuals residing in areas with low mosquito density could experience a significant risk of mosquito bites while attending other locations, such as workplaces, schools, or public green areas. This highlights the need to improve estimates of human exposure to mosquito bites, especially in highly urbanized settings. Although temperature and precipitations are expected to be relatively homogeneous within a single municipality, differences in the spatial resolution of temperature, precipitation, and human density data might lead to potentially biased estimates of the transmission risk. This effect could be especially relevant in areas with strong climatic heterogeneity, such as mountain areas, where, however, the risk of arboviral transmission is consistently low. Finally, the symptom description in this paper is based on notifications, not a clinical study. Cases were classified as confirmed according to the European case definition, but a detailed clinical classification (e.g., mild vs severe dengue) or a critical appraisal of symptoms was not possible.

Nonetheless, our findings provide valuable insights that can be leveraged to enhance awareness of epidemiological risks at the point of care, ultimately increasing effective testing for febrile illnesses during the summer months. The study identifies the symptoms observed in confirmed cases, the most frequent origins of imported infections from endemic area, the time periods of highest importation and local transmission risks. Spatiotemporal risk maps also highlight areas with a higher likelihood of transmission in the future, while suggesting that cases could emerge in specific regions, regardless of prior infection history in those regions. Key actions to prevent and mitigate outbreaks in the future should include improving human case identification in the summer season and raising awareness among health-care workers and the general population on local DENV and CHIKV transmission to support a decrease in delays in detection and response.

## Methods

### Surveillance and control of aedes-borne arboviruses in italy
The human surveillance of arboviral infections in Italy is coordinated by the Ministry of Health, implemented with the technical-scientific know-how of the Italian National Institute of Health (ISS) according to the National Arbovirus Response Plan 2020–2025[39].

In the Italian regionalized health system organization, Regional local health units are in charge of reporting the occurrence and most likely place of exposure of acute human arboviral infections to the National Surveillance System and coordinate activities in the event of health emergencies. Human surveillance of imported and locally acquired cases of DENV and CHIKV is active throughout the year, with case definitions aligned with the EU case definition[40]. Any detection of *Aedes*-borne human infections in Italy (imported or local) is performed by clinicians (GPs or hospitals depending on case severity) who request laboratory confirmation (via PCR or serological tests). Any probable/suspected case as per the EU case definition[41] triggers mandatory reporting within 12 h to the Public Health services and local response within 24 h including case investigation, vector capture, vector control and risk communication activities. In the presence of either confirmed or suspected human cases of arbovirus infection, whether imported or autochthonous, the competent health authority activates the vector control interventions within 24 h of notification. Control interventions are based on disinfestation of the affected area (~200 m radius around the place where the human case presumably was exposed) with insecticides, giving priority to adulticide interventions, both on public land and on private premises, and research and elimination of peri-domestic larval breeding sites, with "door-to-door" inspections of the homes included in the reported area.

If a locally acquired case is suspected, based on the patient interviews, local level active case investigation, proximity vector monitoring (including xenomonitoring) and control are further enhanced. This is why the presence of CHIKV and DENV also in local mosquito pools were confirmed during the observed outbreaks[9,18,42]. Substance of Human Origin (SoHO) safety measures (CHIKV/DENV testing/suspension) are implemented at municipality level[38] upon confirmation of auto-chthonous transmission as well as a 28-day deferral from donation of people who traveled in the affected areas. Additional response measures include national and regional enhanced support for surveillance and active case finding (including voluntary screening campaigns[9]), national referral laboratory activities (including genomic epidemiology), medical entomology and risk communication. Activities conducted by local health authorities include individual case investigations that are not reported at national level. Similarly, control activities aimed at preventing mosquito population growth in the absence of suspected cases are not reported at the national level.

### Epidemiological data
We retrieved data pertaining 1577 laboratory confirmed travel-related cases and 481 autochthonous cases of DENV and CHIKV, notified between 2006 and 2023 to the national database of human surveillance of arboviral infections in Italy hosted by the ISS. These records were obtained after excluding 64 travel-related cases and 10 autochthonous cases, due to data incompleteness or inconsistencies between the dates of symptom onset and notification.

Analyzed variables for each reported case included: pathogen laboratory confirmation, date of symptom onset, date of notification, age, sex, the list of clinical symptoms experienced by the case, classification of cases as travel-related or locally acquired, and geolocation of likely exposure for autochthonous cases and the country of likely exposure as identified during epidemiological interviews. Laboratory data were available only for cases diagnosed from 2013 onwards, encompassing a total of 1193 cases (371 autochthonous and 822 travel-related), with 67% confirmed via PCR and 33% through serological testing.

### Statistical analysis of imported cases
We analyzed the time series of imported cases of both DENV and CHIKV in Italy using a generalized additive model. Generalized Additive Models (GAMs) were preferred over linear models because they accommodate nonlinear effects through smooth functions of covariates. This is particularly useful for modeling travel-related cases, which are expected to exhibit seasonal patterns rather than a simple, monotonic relationship with the months. Furthermore, GAMs mitigate overfitting by employing penalized regression splines, resulting in a more robust and interpretable model. Specifically, we considered the number of imported cases per month, aggregated by the date of symptom onset, as the response variable and assumed that it follows a Poisson distribution. A log link function was considered. The model incorporates two qualitative covariates: the disease (dengue or chikungunya), and whether the case importation occurred during travel restrictions put in place during the COVID-19 crisis (i.e., symptom onset between February 2020 and December 2020)[43]. Regarding quantitative covariates, the year and month of symptom onset were modeled as penalized cubic splines and penalized cyclic cubic splines, respectively. To explore the relationships between each disease and both the month and year of importation, interactions between these temporal variables and the disease were considered, resulting in four

smoothing functions. Finally, an autoregressive term of order two was included in the model to control for temporal autocorrelation. The analysis was carried out in a frequentist framework using the function gamm implemented in the R package mgcv (R Project for Statistical Computing, software version 4.3.2).

Following a similar approach, we investigated possible temporal changes in the delay associated with the notification of imported cases to the central health authority, defined as the time between the symptom onset of cases and their reporting to the National Surveillance System through the ISS surveillance platform. In this case, we employed a Generalized Additive Mixed Model (GAMM), by assuming that the notification delay follows a Negative Binomial distribution with log link. Whether the case was dengue or chikungunya was included as a qualitative covariate. The year and month of symptom onset were modeled as penalized cubic splines and penalized cyclic cubic splines, respectively. The interaction between these terms and the disease was accounted for as in the GAM. The region of importation was considered as a random effect.

We finally investigated whether the notification delay differed between autochthonous and imported cases by means of Generalized Linear Mixed Models (GLMM), assuming a Negative Binomial distribution with log link. To do this, we included autochthonous cases in our analysis and focused solely on regions and years where autochthonous transmission was documented (chikungunya: 2007 in Emilia-Romagna, 2017 in Lazio and Calabria; dengue: 2020 in Veneto, 2023 in Lazio and Lombardy). Two categories of autochthonous cases were considered: those with symptom onset preceding the outbreak detection and those occurring afterward. Considered covariates included the disease and case classification (i.e., imported, autochthonous preceding the outbreak detection, autochthonous following the outbreak detection); the outbreak was considered as a random effect. The analysis was carried out in a frequentist framework using the function glmer.nb implemented in the R package lme4 (R Project for Statistical Computing, software version 4.3.2). More details are provided in the Supplementary Information.

### Transmissibility from surveillance of human cases

We estimated the net reproduction number ($R_t$) associated with each arboviral outbreak occurred in Italy between 2006 and 2023, by applying a consolidated Bayesian approach[44,45]. Estimates of $R_t$ were obtained using a Markov Chain Monte Carlo (MCMC) sampling method and applying the renewal equation to the time series of symptomatic confirmed cases by date of symptom onset. This probabilistic approach accounts for uncertainty in $R_t$ estimates, making it particularly well-suited for real-time epidemic analysis[8,9]. We assumed a Gamma-distributed generation time with a mean of 12.4 days and a variance of 18.5 for CHIKV[14], and with a mean of 18 days and a variance of 66 for DENV[8]. The analysis was performed using codes widely adopted for estimating the net reproduction number available at https://github.com/majelli/Rt. More details are provided in the Supplementary Information.

### Assessment of onward transmission risks from entomological and climate data

We assessed the potential risk of onward transmission for chikungunya and dengue at the national level during the outbreak years: 2007, 2017, 2020, and 2023. To this aim, we leveraged a model recently developed to estimate the spatiotemporal abundance of *Aedes spp.* mosquitoes and the consequent risk of autochthonous arboviral transmission at detailed spatiotemporal scales[26]. The model was extended to incorporate year-specific daily temperature records, account for more refined data on human density (at a resolution of 100 m × 100 m), and provide for each year of interest (1) daily estimates of the CHIKV and DENV reproduction number $R_0$ (i.e., the average number of secondary human infections arising from a primary human infector in a fully susceptible population), (2) the likelihood of onward transmission, defined as the probability of experiencing local transmission chains after case importation, and (3) the duration of epidemic risks (defined as the number of consecutive days in which $R_0$ exceeds the epidemic threshold of 1). Specifically, the absolute density of adult mosquitoes per hectare on a given day was modeled as a logistic function of the mean temperature observed in the preceding days, where the maximum is defined as proportional to a location-specific climatic suitability index ($\sigma_i$)[26]. This approach assumes that seasonal variations in mosquito abundance are primarily driven by the persistence of favorable temperature conditions throughout the mosquito lifecycle and that higher abundances are expected in locations associated with greater climatic suitability, as described by the following equation for any day $d$ and location $i$:

$$N_v(i, d) = \frac{\alpha \sigma_i}{\left(1 + e^{-k(\widetilde{T}(d,i,w) - T_0)}\right)} \tag{1}$$

where $\widetilde{T}(d, i, w)$ represents the average temperature recorded at location $i$ in the $w$ days that precede $d$, $T_0$ and $k$ are the midpoint and steepness parameters of the logistic function, and $\alpha$ is a scaling factor that accounts for variations in trap efficiency across mosquito capture data used for model calibration, as well as consequent rescaling factors to convert estimated captures into per-hectare abundance. The climate suitability index ($\sigma_i$) was assumed to be driven by temperature and precipitation observed in location $i$ as in Zardini et al.[26]. Overall, the model was informed by georeferenced presence-absence records of *Ae. albopictus* across 4372 locations in Europe and 300 time-series of female adult mosquitoes collected between 2007 and 2018. The resulting estimates of vector abundance were combined with human density retrieved from the WorldPop data[46] to assess arbovirus transmission potential. Specifically, for each location and year of interest, we calculated the basic reproduction number ($R_0$) and the likelihood of onward transmission using a Susceptible, Exposed, Infectious, Recovered (SEIR) host and Susceptible, Exposed, Infectious (SEI) vector model. Both measures were computed following standard approaches as proportional to the vector-to-host ratio ($N_v(i, d)/N_h(i)$), while incorporating disease- and vector-specific parameters obtained from the literature. More details are provided in the Supplementary Information.

We compared estimates of $R_0$ resulting by assuming *Ae. albopictus* as the only vector for arboviral transmission in Italy with $R_t$ values estimated from human cases identified during the different outbreaks. To do this, we combined $R_0$ estimates associated with geographical patches characterized by a population density greater than 10 residents per hectare and falling within a square of 900 m × 900 m centered on the site of likely exposure of each autochthonous case. The square dimension was determined based on the analysis of past chikungunya and dengue epidemics, suggesting that most transmission events for both diseases typically occur within a 900 m radius[8,14,45]. Daily temperature records were retrieved at a spatial resolution of 0.1° × 0.1° from the E-OBS dataset, which is a collection of high-resolution gridded climate data covering Europe and part of the Copernicus Climate Change Service[47]. Human density data were obtained at a spatial resolution of 100 m × 100 m from the WorldPop data[46]. The analysis was performed using codes developed ad-hoc by our team in the programming language C, available at https://zenodo.org/records/10203374. Maps were created using QGIS software version 3.30.2. Administrative boundaries of the Italian regions were retrieved from the Italian Institute of Statistics[48]. More details are provided in the Supplementary Information.

### Reporting summary

Further information on research design is available in the Nature Portfolio Reporting Summary linked to this article.

## Data availability

The data contain confidential information, and public data deposition is not permitted. Due to the sensitive nature of the data, raw data can only be made available by the Istituto Superiore di Sanità (Italian National Institute of Health) through a data-sharing agreement directly with the user (contact mail: sorveglianza.arbovirosi@iss.it).Aggregated data are available at https://www.epicentro.iss.it/arbovirosi/dashboard and https://www.epicentro.iss.it/arbovirosi/bollettini.

## Code availability

The statistical analysis was performed using R statistical software version 4.2.2 2022 (R Project for Statistical Computing). The code is available on Zenodo at https://zenodo.org/records/15100955. The analysis of reproduction numbers and the potential risk of onward transmission was performed using codes developed ad-hoc by our team in the programming language C, available at https://zenodo.org/records/10203374, and codes widely adopted for estimating the net reproduction number available at https://github.com/majelli/Rt.

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

## Acknowledgements

This research was supported by European Union funding under NextGenerationEU-MUR PNRR Extended Partnership Initiative on Emerging Infectious Diseases (project no. PE00000007 INF-ACT CUP H93C22000640007) (F.Me., M.M., A.Z., E.A.F., P.Po., S.M., A.B., A.T.P., P.Pe., F.Ri., M.D.L., G.V., and C.F.).

## Author contributions

P.Po., P.Pe., S.M., and F.Ri. conceived and supervised the study. F.Me., M.M., A.G., M.D.M., and A.B. performed the analyses. F.Me., M.M., P.Po., and F.Ri. wrote the first draft of the manuscript. M.D.M., A.B., A.D.M., Gio.M., F.V., L.V., F.Ru., F.L., Giu.M., E.R.D.G., F.Sc., F.Mo., M.T., F.Z., C.D.L., and P.S. collected the data reported in the manuscript. M.M., M.D.M., A.B., and P.Po. directly accessed and verified the data. F.Me., M.M., M.D.M., A.B., A.Z., A.G., A.D.M., Gio.M., F.V., L.V., F.Ru., F.F., F.Ma., A.T.P., E.A.F., M.D.L., F.Se., L.T., G.V., C.F., C.M., F.L., Giu.M., E.R.D.G., F.Sc., F.Mo., M.T., F.Z., C.D.L., P.S., P.Po., P.Pe., S.M., and F.Ri. contributed to interpreting the results; and read, reviewed, and approved the final version and the submission of the manuscript.

## Competing interests

The authors declare no competing interests.

## Ethics approval

This study was conducted using epidemiological data from the Italian national integrated arbovirus surveillance routinely collected and analyzed within the mandate of the ISS; therefore, no ethical approval was necessary.

## Additional information

## Dengue risk assessment working group

**Emmanouil A. Fotakis**[1,2], **Marco Di Luca**[2], **Francesco Severini**[2], **Luciano Toma**[2], **Giulietta Venturi**[2], **Claudia Fortuna**[2], **Christina Merakou**[2], **Francesco Lucia**[3], **Giulio Matteo**[4], **Esther Rita De Gioia**[4], **Francesco Scovenna**[6], **Federica Morani**[6], **Michele Tonon**[7], **Francesca Zanella**[7], **Claudio De Liberato**[9] & **Paola Scaramozzino**[9]

[9]Istituto Zooprofilattico Sperimentale del Lazio e della Toscana M. Aleandri, Rome, Italy.

