## [Transparent Peer Review file · Nature Communications]

Risk assessment and perspectives of local transmission of chikungunya and dengue in Italy, a European forerunner

Corresponding Author: Dr Piero Poletti

Version 0:

Reviewer comments:

Reviewer #1

(Remarks to the Author)

This is a timely paper highlighting the current and future risks of dengue and chikungunya outbreaks in Italy. The work is of significance considering the increasing threat of arboviral infection transmission in Italy and in Southern Europe. The introduction is adequate and sets the scene for analyses. I suggest that the authors make more use of the rich data set that is available to them and also provide more detail on the public health response to both the importation and local transmission of DENV and CHIKV. Specifically can the authors add sections to both the methodology and results and clarify the following:

1. The national reporting system - who reports cases of DENV and CHIKV? Is this done by physicians and hospitals and laboratories or by a combination? Is reporting mandatory? In the "results" section you report on 1577 travel related cases of DENV and CHIKV but few details are provided on these cases? There is some information on notification delays but no details on the countries of acquisition of the cases, age and sex of the cases, testing method used, severity of clinical symptoms or outcomes. Imported cases of arboviral infection mean viraemic individuals that are key for outbreaks so some details on these cases are essential. Some details should also be provided on the index cases in the outbreaks.
2. Another key component of DENV and CHIKV transmission is the entomological situation and environmental conditions. This is alluded to in the discussion but not addressed. The title suggests future "perspectives" but no mention is made of changing climate, higher temperatures and increased precipitation and how these factors will have an effect on the geographic range and vectorial capacity of *Aedes albopictus* - can the authors address this deficit and also bring some references to standardised climate scenarios. What methods are currently used for mosquito control in urban Italy? Can lessons be learned from other cities that have successfully combated DENV?
3. The public health measures for combating *Aedes* and possible DENV and CHIKV infections are only referred to in passing. Can the authors provide details? What is being done now? What should be improved on? In their conclusion the authors suggest "improving human case identification" but can they say how this should be done as up to 80% of DENV cases are asymptomatic or have only mild symptoms. Should there be rapid testing of returning travellers? Should airports highlight risk of DENV post travel? Should rapid tests be available in community pharmacies and in general practice? Can national surveillance be strengthened? Should public health communication campaigns be done during risk periods? Some practical sections would clarify.

Reviewer: Patricia Schlagenhauf

Reviewer #2

(Remarks to the Author)

Menagale F. et al., in this article, present a brilliant and timely study assessing the spatiotemporal patterns of chikungunya and dengue importation in Italy. By leveraging human, entomological, and climate data, the authors calculate reproduction numbers for six local outbreaks and map current transmission risks. The topic is highly relevant in the context of the increasing threat of arboviral diseases in Europe, and the study provides valuable insights into the factors driving local transmission in a European setting.

The figures and results are presented in a highly elegant and visually appealing manner, effectively communicating the key findings of the study. The integration of diverse datasets and methodologies adds significant value to the research.

However, there are areas that could further enhance the quality and impact of this work:

Methods Section: While the study is methodologically robust, the methods section would benefit from more detailed descriptions of the analytical approaches used. This includes clarifying how the spatiotemporal patterns were analyzed, the specific models or assumptions underlying the reproduction number calculations, and the methodology for integrating human, entomological, and climate data.

Data Availability: A dedicated data availability section is currently missing. To improve transparency and reproducibility, it is essential to provide access to input R codes, processed datasets, and relevant outputs in a public repository. This would enable other researchers to validate the findings and build upon this important work.

Reproducibility and Accessibility: Sharing the computational tools and datasets would also align with best practices in scientific research, allowing for broader applicability and encouraging future studies to leverage this framework for risk assessment in other regions.

In summary, while this study is undoubtedly a valuable contribution to the field, addressing these points would further enhance its scientific rigor and accessibility, ensuring its long-term impact on the understanding and mitigation of arboviral transmission risks in Europe.

Reviewer #3

(Remarks to the Author)

This article is of great interest and possibly well conducted. However, it is extremely hard to judge it because the Methods section did not explain how the analyses were performed sufficiently well. The manuscript is also challenging to read because some important information has been left out, or misplaced, throughout and many details that are not needed are in the main text and obstruct the flow. I suggest a thorough rewrite of the entire manuscript, so that I can better assess it. Following are my suggestions in each section/paragraph as I went through the manuscript.

Abstract

Please expand on how exactly you analyze the data.

The words "assessed" and "leveraging" could be switched for more precise wording.

Introduction

Second half of third paragraph seems off topic, starting at the fourth sentence on line 67. The first half discusses the recent outbreaks of dengue, but then the second half start a discussion about the different levels of government involved. While the first half is interesting and fits neatly in this position in the Introduction, the second is not very interesting for most readers and belongs more in the Material and Methods, perhaps in a section about description of the study area. I recommend merging the first half of the third paragraph, down to line 67 at the end of the third sentence, with the second paragraph, and either delete the second half or move it to the Material and Methods.

The fourth and fifth paragraphs give little detail about the study framework or the analyses performed in this study. They both describe certain programs and systems in place at various levels of government to investigate disease outbreaks. While this may be interesting to some readers, the space in these two paragraphs would be much better used by describing the datasets used, in terms of where and when and which types of variables were collected, and by explaining the type of analyses they performed in this study and why, and their hypotheses. I would delete the fourth and fifth paragraphs, and expand the last paragraph for a much more interesting and informative introduction.

Results

Please choose a syntax form and keep it throughout for DENV/CHIKV or dengue/chikungunya.

No need to refer to the Method section at line 147 because this is a given.

Describe in figure captions which software was used to generate maps, where appropriate.

I think Figure 5 could be removed or placed in supplementary material.
From which results do you draw the conclusion at lines 162-164?

Figure 6 seems to contain two unrelated figures. I would put panels A and B in its own figure, and C, D, E and F in a separate one. If number of figures is restricted by the journal, deleting Figure 5 should open up space to divide Figure 6 into two.

Discussion

Discussion is currently about 3 times the length of the Introduction. Many elements of the Discussion, particularly in the few earlier paragraphs, would best be presented in the Introduction. The Discussion should mostly be where results on this study are discussed in light of past results from previous studies, except a paragraph at the beginning that serves as a

reminder of the study objectives, and a paragraph at the end that serves as a summary of the study and perhaps also as a take home message. These are standard guidelines, that may not be applicable to all articles, but following them more closely here could really benefit the article as a whole and make the message flow better for the reader.

Also, remember to always write the entire name of abbreviations at first mention, such as WHO and ISS.

Methods

In the section starting at line 331, state the softwares and version you used for the generalized additive mixed models. Why did you use generalized additive mixed models, instead of other types of generalized mixed models? Which variables were used as random-effects variables? For example, "the interaction between these terms and the disease was also considered" is way too vague. How were the interaction terms considered? Did you include all interaction terms in the models as fixed-effects variables? Did you test correlation among them and their respective source variables?

At the paragraph starting at line 351, you need to add more details on how you ran this analysis. How did you investigate the comparison between autochthonous and imported cases? Which software, which software version, what analysis type, and if modelling, which type of model did you use? These are more important than most of what is currently written in this paragraph. If space is limited, most of the current paragraph could be relegated to the Appendix, as already referred to in the same paragraph.

At the paragraph starting at line 360, describe the consolidated Bayesian approach in a more precise way, and explain why you use this approach? Also, state the software, version and functions you used. The section is small and should have more details in the main text and less in the Appendix.

In the prospective analysis starting at line 367, very few details were given by the authors, which makes it very difficult to assess how the analysis was conducted and the extent of its limitations. While I cannot fully assess how this analysis was done and therefore if relevant, a large difference in resolution among the various explanatory variables used may be a source of spatial biases. Also, data use to estimate *Aedes* spp. abundance was not described. Did you use mosquito capture and arbovirus testing data, and of which species exactly? If this analysis relies only on human cases, describe how the estimates are still accurate and meaningful, and otherwise I believe this limitation should be noted in the Discussion. Daily temperature records at a resolution of 0.1 x 0.1 degrees is very coarse. You may obtain land surface temperature raster files at a 1km x 1km resolution from USGS. Did you also consider precipitation, which are very important to predict mosquito habitat? Global Human Settlement Layer is more a land use dataset, showing built-up density, which is somewhat correlated with human population density. If the authors need actual human population density, Worldpop might be more appropriate.

The paragraph starting at line 391 is very important and useful, but I believe these details should be expanded on and added throughout the Methods section.

Version 1:

Reviewer comments:

Reviewer #1

(Remarks to the Author)

The authors have addressed my reviewer comments in detail and the resulting revised manuscript is clear and is a very useful paper. I recommend acceptance.

Reviewer: Patricia Schlagenhauf

(Remarks on code availability)

Reviewer #2

(Remarks to the Author)

The authors have addressed all the reviewers' comments and incorporated the suggested changes. I believe the revised manuscript now meets the requirements and is suitable for publication.

(Remarks on code availability)

The authors have addressed all the reviewers' comments and incorporated the suggested changes. I believe the revised manuscript now meets the requirements and is suitable for publication.

Reviewer #3

(Remarks to the Author)

All my comments have been properly addressed, thank you.

(Remarks on code availability)

RESPONSE TO THE REVIEWERS' COMMENTS

Reviewer #1 (Remarks to the Author):

This is a timely paper highlighting the current and future risks of dengue and chikungunya outbreaks in Italy. The work is of significance considering the increasing threat of arboviral infection transmission in Italy and in Southern Europe. The introduction is adequate and sets the scene for analyses. I suggest that the authors make more use of the rich data set that is available to them and also provide more detail on the public health response to both the importation and local transmission of DENV and CHIKV.

We would like to thank the Reviewer for taking time to evaluate our manuscript, for appreciating our work, and for the valuable comments provided. In this revised version of our manuscript, we have expanded our analysis of imported cases and symptoms in confirmed cases while also providing further details on the analyzed data and the public health measures implemented to mitigate the risks of *Aedes*-borne transmission in Italy. Below, we present a detailed, point-by-point response to all comments raised by the Reviewer.

Specifically can the authors add sections to both the methodology and results and clarify the following:

1. The national reporting system - who reports cases of DENV and CHIKV? Is this done by physicians and hospitals and laboratories or by a combination? Is reporting mandatory? In the "results" section you report on 1577 travel related cases of DENV and CHIKV but few details are provided on these cases? There is some information on notification delays but no details on the countries of acquisition of the cases, age and sex of the cases, testing method used, severity of clinical symptoms or outcomes. Imported cases of arboviral infection mean viraemic individuals that are key for outbreaks so some details on these cases are essential. Some details should also be provided on the index cases in the outbreaks.

We thank the Reviewer for the valuable suggestions. In response, we have added a new subsection in the *Methods* describing the surveillance and control of *Aedes*-borne arboviruses in Italy, as well as another subsection in the *Results* detailing the characteristics of ascertained cases, including information on sex, age, most common symptoms, infection-related fatalities, and the country of origin for travel-related cases. Further details on the surveillance system are now provided in the Supplementary Information under the new section, "*Surveillance and Control of CHIKV and DENV Infections in Italy*". This section includes a new table (Table S1) summarizing the characteristics of travel-related and autochthonous cases, along with two maps illustrating the reported countries of exposure for imported DENV and CHIKV cases (Fig. S1). Accordingly, we have also revised the *Discussion* to highlight key insights from these findings. For the Reviewer's convenience, we provide below excerpts from the revised main text addressing all the points raised, along with the new figures and table.

“In the Italian regionalized health system organization, Regional local health units are in charge of reporting the occurrence and most likely place of exposure of acute human arboviral infections to the National Surveillance System and coordinate activities in the event of health emergencies. Human surveillance of imported and locally acquired cases of DENV and CHIKV is active throughout the year, with case definitions aligned with the EU case definition⁴⁰. Any detection of *Aedes*-borne human infections in Italy (imported or local) is performed by clinicians (GPs or hospitals depending on case severity) who request laboratory confirmation (via PCR or serological tests). Any probable/suspected case as per the EU case definition⁴¹ triggers mandatory reporting within 12 hours to the Public Health services and local response within 24 hours including case investigation, vector capture, vector control and risk communication activities.”

[...]

“Between 2006 and 2023, a total of 1,577 travel-related laboratory-confirmed cases of DENV and CHIKV infections were reported to the Italian national surveillance system (1,435 DENV infections and 142 CHIKV infections). The median age of travel-related cases was similar for the two diseases (36 years, IQR: 27-48; 39 years, IQR: 30-54, for DENV and CHIKV, respectively). Male to female ratios for the two diseases were 1.3 (males: 56.3% for CHIKV vs 55.7% for DENV). (see Supplementary Table S1). The information on the country of exposure was reported for 98.6% of travel related cases, encompassing 87 countries of exposure (see Supplementary Fig. S1). The most frequently reported countries of exposure for DENV infections diagnosed in Italy were Thailand, Cuba, India, and Maldives, accounting for 43.0% of imported DENV cases while the most frequently reported countries of exposure for CHIKV infections diagnosed in Italy were India, Dominican Republic, Brazil, and Thailand, accounting for 47.9% of CHIKV imported cases. During the same period, 481 autochthonous laboratory-confirmed cases of CHIKV (93 cases) and DENV (388 cases) were reported. The median age of autochthonous cases was 57 (IQR: 38-69) and 58 (IQR: 42-72) for DENV and CHIKV respectively. Similar male to female ratios were found in the two diseases (males: 52.7% for DENV vs 50.0% for CHIKV).”

[...]

“Overall, most common symptoms reported by DENV cases were fever (reported by 499 cases), arthralgias (484), skin rash (310), asthenia (362), and meningoencephalitis (5). Any type of hemorrhagic symptoms including bleeding from gums or nose and petechiae were rare (23 cases, 1.5%). The outcome was favorable for all cases, with no reported fatalities. Most common symptoms reported by CHIKV cases were fever (1,424), arthralgias (956), skin rash (708), asthenia (1,111), and meningoencephalitis (10). Out of all diagnosed cases, only two fatalities were reported, with the remaining patients achieving recovery.”

[...]

“Reported countries of exposure to DENV and CHIKV infections were in line with those documented by the European Centre for Disease Prevention and Control (ECDC) for travel-related CHIKV and DENV cases ascertained in the European Union from 2019 to 2023^{24,25}.”

[...]

“our results show that control of local outbreaks, defined as R_t below 1, was mostly reached within one generation time following the outbreak detection (index case). This finding underscores the importance of early identification of autochthonous cases, as initial detection often occurs weeks after introduction, with a suspected primary case of local transmission identified in only 3 out of 6 outbreaks analyzed^{10,12,16,28}. A recent

review highlights the challenge of detecting primary imported DENV/CHIKV cases in Europe, with median delays of over 35 days between the symptom onset of the primary case and the diagnosis of the index case¹⁰.”

[...]

“The description of clinical symptoms reported here, combined with the key seasonal timing and travel history of imported cases reported in the past, may increase awareness and enhance the probability of recognizing symptoms of arboviral infections, particularly during the season of high vector activity”

[...]

“Analyzed variables for each reported case included: pathogen laboratory confirmation, date of symptom onset, date of notification, age, sex, the list of clinical symptoms experienced by the case, classification of cases as travel-related or locally acquired, and geolocation of likely exposure for autochthonous cases and the country of likely exposure as identified during epidemiological interviews. Laboratory data were available only for cases diagnosed from 2013 onwards, encompassing a total of 1,193 cases (371 autochthonous and 822 travel-related), with 67% confirmed via PCR and 33% through serological testing.”

Table S1. Demographic and clinical characteristics of DENV and CHIKV laboratory-confirmed infections notified to the Italian National Surveillance system (n=2,058), Italy, 2006 and 2023.

	CHIKV (n=530)		DENV (n=1,528)	
	n	%	n	%
Age group (year)				
<20	42	8.0	140	9.2
20-39	111	21.2	737	48.5
40-59	165	31.5	485	31.9
60+	206	39.3	157	10.3
Sex				
Female	256	48.3	679	44.5
Male	274	51.7	848	55.5
Symptom				
Fever	499	94.1	1424	93.2
Arthralgias	484	91.3	956	62.6
Skin rash	310	58.5	708	46.3
Asthenia	362	68.3	1111	72.7
Meningoencephalitis	5	0.9	10	0.6
Diarrhea	33	6.2	65	4.2
Vomit	32	6.0	39	2.6
Itching	38	7.2	15	1.0
Thrombocytopenia	1	0.2	31	2.0
Nausea	17	3.2	24	1.6
Myalgia	24	4.5	1	0.1
Headache	22	4.2	0	0.0
Photophobia	7	1.3	2	0.1
Conjunctivitis	15	2.8	24	1.6

Haemorrhagic symptoms	2	0.4	23	1.5
Lymphadenopathy	3	0.6	6	0.4
Edema	7	1.3	2	0.1
Arthritis	2	0.4	0	0.0
Other neurological symptoms	3	0.6	3	0.2
Hepatosplenomegaly	1	0.2	3	0.2
Syncope	0	0.0	4	0.3
Gastric/Abdominal pain	0	0.0	13	0.9
Other symptoms	4	0.7	33	2.2

Fig. S1. Countries of exposure reported by imported DENV and CHIKV cases.

2. Another key component of DENV and CHIKV transmission is the entomological situation and environmental conditions. This is alluded to in the discussion but not addressed. The title suggests future "perspectives", but no mention is made of changing climate, higher temperatures and increased precipitation and how these factors will have an effect on the geographic range and vectorial capacity of *Aedes*

albopictus - can the authors address this deficit and also bring some references to standardised climate scenarios.

We apologize for the lack of a proper discussion on these points in the previous version of our manuscript. We fully agree with the Reviewer that temporal changes in environmental conditions, especially temperature, are key drivers of arboviral transmission risks. Our model, based on entomological and climate data, explicitly incorporates year-specific daily temperature records to retrospectively assess potential trends in transmission risk from 2007 to 2023. The resulting estimates, shown in Panels A and B of Figure 6 (Figure 5 in the revised version of the manuscript), indicate that the risk of onward transmission for CHIKV and DENV has not exhibited a clear trend over the last 17 years. In contrast, season-specific conditions resulted predominant over the period considered, with significant inter-annual variations in transmission risk observed for both infections, particularly in the northwest and central regions of Italy. On the other hand, we provide evidence that CHIKV and DENV transmission is possible in urban areas of Italy, that transmission risks in southern regions extend into late autumn, and that outbreaks can persist for months even when the net reproduction number (R_t) drops below the epidemic threshold.

Although we did not assess future scenarios, we believe that both risk maps—calibrated on entomological data—and the epidemiological analysis of travel-related and autochthonous human cases over almost two decades provide a key perspective on possible future scenarios for other European countries currently experiencing the introduction or increasing colonization of *Ae. albopictus*. While we now reference previous results from our group indicating that the geographic expansion of *Ae. albopictus* is expected over the next decades across large portions of Europe (Zardini et al., *The Lancet Planetary Health*, 2024), we acknowledge that this study does not explore how changes in climate, urbanization, vector capacity, and inter-country mobility may influence future transmission risks.

To address the Reviewer's comment, we have expanded the *Discussion* as follows:

“Even if the risk of onward transmission ($R_0 > 1$) for CHIKV and DENV does not show a clear yearly trend since 2007, we can expect an increase in the frequency of local DENV transmission events due to clear increases in the importation trends. Based on concurrent evidence from human and entomological surveillance, we expect that local DENV transmission events will be more frequent than CHIKV local transmission events. However, once initiated, we expect onward transmission of CHIKV to be more effective and determine larger outbreaks due to the combination of a higher R_0 and a shorter generation time. Although season-specific effects remained crucial in the analyzed outbreaks, we presented evidence that CHIKV and DENV transmission in urban areas of Italy is possible and that transmission risks in southern regions may extend into late autumn. We also showed that, in Italy's long, warm autumns, outbreaks can persist for months even when R_t drops below the epidemic threshold, with transmission typically ending as temperatures decrease. This has resulted in relatively large CHIKV and DENV outbreaks compared to other EU/EEA countries. As suggested by 2024 data currently under consolidation^{20,38}, epidemic risks are increasing, with several concomitant local clusters reported, including one with over

100 locally acquired DENV infections⁹. As the geographic expansion of *Ae. albopictus* is expected across large portions of Europe²⁶, further efforts are needed to quantify how changes in climate, urbanization, vector capacity, and inter-country mobility may influence future arboviral transmission risks.”

What methods are currently used for mosquito control in urban Italy? Can lessons be learned from other cities that have successfully combatted DENV?

As mentioned above, we revised our manuscript to include a detailed description of surveillance and vector control measures in place in Italy. In Italy, Regional Local Health Units are responsible for coordinating activities in response to health emergencies. In the presence of either confirmed or suspected human cases of arbovirus infection — whether imported or autochthonous — the competent health authority must activate vector control interventions within 24 hours of notification. These control measures include disinfestation of the affected area (approximately a 200-meter radius around the suspected exposure site) using insecticides, prioritizing adulticide interventions on both public and private land, and door-to-door inspections to identify and eliminate peri-domestic larval breeding sites within the affected area. Control activities aimed at preventing mosquito population growth in the absence of suspected cases are not reported at the national level. All this information is provided in the new section entitled “*Surveillance and Control of Aedes-borne Arboviruses in Italy*” in the main text and further detailed in the Supplementary Information.

The protocol for outbreak management is standardized, with public health measures and control interventions applied uniformly, regardless of the size of urban settlements or level of urbanization, following the National Arbovirus Response Plan 2020–2025. To better clarify and discuss the point, we expanded the main text as follows:

“Once detected, the protocol for outbreak management is standardized and a package of measures and control interventions implemented regardless of the size of urban settlements or level of urbanization. However, a higher difficulty in controlling transmission was observed in large urban settings (e.g., Rome) where the risk is more heterogeneous than in smaller cities, a higher frequency of importation is expected within the same season, and multiple transmission events have been observed to occur simultaneously. Key factors that may limit the effectiveness of control interventions include difficulties in accessing private properties and the resistance of *Ae. albopictus* to pyrethroid insecticides³⁰⁻³². Additionally, in highly populated cities, human mobility complicates identifying exposure sites and areas of sustained transmission, making it harder to identify at-risk populations and implement effective vector monitoring and control activities. This notwithstanding, our results show that control of local outbreaks, defined as R_t below 1, was mostly reached within one generation time following the outbreak detection (index case). This finding underscores the importance of early identification of autochthonous cases, as initial detection often occurs weeks after introduction, with a suspected primary case of local transmission identified in only 3 out of 6 outbreaks analyzed^{10,12,16,28}.”

[...]

“Similarly, control activities aimed at preventing mosquito population growth in the absence of suspected cases are not reported at the national level.”

3. The public health measures for combating Aedes and possible DENV and CHIKV infections are only referred to in passing. Can the authors provide details? What is being done now? What should be improved on? In their conclusion the authors suggest "improving human case identification" but can they say how this should be done as up to 80% of DENV cases are asymptomatic or have only mild symptoms. Should there be rapid testing of returning travellers? Should airports highlight risk of DENV post travel? Should rapid tests be available in community pharmacies and in general practice? Can national surveillance be strengthened? Should public health communication campaigns be done during risk periods? Some practical sections would clarify.

Reviewer: Patricia Schlagenhauf

As noted above, we have added a detailed description of the measures implemented to mitigate mosquito proliferation and control DENV and CHIKV outbreaks in both the main text and the Supplementary Information. Specifically, the following text has been added to the *Methods* Section:

“In the presence of either confirmed or suspected human cases of arbovirus infection, whether imported or autochthonous, the competent health authority activates the vector control interventions within 24 hours of notification. Control interventions are based on disinfestation of the affected area (approximately 200 meters radius around the place where the human case presumably was exposed) with insecticides, giving priority to adulticide interventions, both on public land and on private premises, and research and elimination of peri-domestic larval breeding sites, with "door-to-door" inspections of the homes included in the reported area.

If a locally acquired case is suspected, based on the patient interviews, local level active case investigation, proximity vector monitoring (including xenomonitoring) and control are further enhanced. This is why the presence of CHIKV and DENV also in local mosquito pools were confirmed during the observed outbreaks^{9,18,42}. Substance of Human Origin (SoHO) safety measures (CHIKV/DENV testing/suspension) are implemented at municipality level³⁸ upon confirmation of autochthonous transmission as well as a 28-day deferral from donation of people who travelled in the affected areas. Additional response measures include national and regional enhanced support for surveillance and active case finding (including voluntary screening campaigns⁹), national referral laboratory activities (including genomic epidemiology), medical entomology and risk communication.”

Additionally, in response to the reviewer's comment, we have expanded the *Discussion* on the challenges of timely case identification and the observed symptomatic ratio of dengue in both endemic and non-endemic countries. We suggest that integrating wastewater surveillance could help bridge the testing gap caused by asymptomatic infections, improving risk monitoring and containment of the spatial spread of ongoing outbreaks. The revised main text now reads as follows:

“A recent review highlights the challenge of detecting primary imported DENV/CHIKV cases in Europe, with median delays of over 35 days between the symptom onset of the primary case and the diagnosis of the index case¹⁰.

In Italy, detection is predominantly limited to symptomatic cases. Serological surveys after the 2007 Italian CHIKV outbreak estimated a symptomatic ratio of 82% or more^{33,34}, but similar data for DENV in non-endemic settings is lacking. Available data from endemic regions suggest a symptomatic ratio of around 20%, while voluntary screenings during the 2023 DENV outbreak in Lombardy found a ratio of 80%⁸. Further studies are needed to determine whether this observation results from under-detection or if the clinical presentation differs between populations rarely exposed to DENV and those with a long history of endemic exposure. In temperate regions, lack of awareness of physicians on *Aedes*-borne arboviral diseases may contribute to an under-diagnosis and in delaying the detection of local outbreaks. The description of clinical symptoms reported here, combined with the key seasonal timing and travel history of imported cases reported in the past, may increase awareness and enhance the probability of recognizing symptoms of arboviral infections, particularly during the season of high vector activity. Given the challenges in case identification¹⁰, and recent advancements in the detection of arboviruses in wastewater samples^{35,36}, wastewater surveillance might become a complementary tool to overcome the limitations posed by asymptomatic and other non-recognized infections. Airplane wastewater monitoring could help identify periods of highest importation risk³⁷, while community-level wastewater surveillance may assist in detecting potential viral spread³⁵, enhancing early warning capabilities triggering epidemiological investigations and containment efforts.”

Reviewer #2 (Remarks to the Author):

Menagale F. et al., in this article, present a brilliant and timely study assessing the spatiotemporal patterns of chikungunya and dengue importation in Italy. By leveraging human, entomological, and climate data, the authors calculate reproduction numbers for six local outbreaks and map current transmission risks. The topic is highly relevant in the context of the increasing threat of arboviral diseases in Europe, and the study provides valuable insights into the factors driving local transmission in a European setting.

The figures and results are presented in a highly elegant and visually appealing manner, effectively communicating the key findings of the study. The integration of diverse datasets and methodologies adds significant value to the research.

However, there are areas that could further enhance the quality and impact of this work:

We would like to thank the reviewer for the time taken to evaluate our manuscript, for the thoughtful words of appreciation for our work, and for the useful comments provided. Here, we provide a detailed point-to-point response to all the comments raised by the Reviewer.

Methods Section: While the study is methodologically robust, the methods section would benefit from more detailed descriptions of the analytical approaches used. This includes clarifying how the spatiotemporal patterns were analyzed, the specific models or assumptions underlying the reproduction number calculations, and the methodology for integrating human, entomological, and climate data.

Following the reviewer's suggestion, we have expanded the description of the methods used to assess transmission risks by leveraging 1) human case surveillance data and 2) a transmission model informed by entomological and climate data. This includes clearer statements on the modeling assumptions and an overview of the methodological approach underlying our estimates of the vector population abundance, the basic reproduction number (R_0), and the likelihood of onward transmission. Additional details are also provided in the Supplementary Information. Finally, we have added a new paragraph at the end of the *Introduction* to clarify how different data sources and models were used to investigate arboviral transmission risks in Italy. The revised text now reads as follows:

INTRODUCTION: "We applied generalized additive models to the time series of travel-related cases to highlight spatiotemporal risks of disease importation and to identify temporal changes in notification delays. We estimated and compared the daily reproduction number for CHIKV and DENV, using consolidated time series of symptomatic cases for each outbreak and a model informed by mosquito capture data, incorporating season-specific temperature data to account for climatic factors."

METHODS: “We estimated the net reproduction number (R_t) associated with each arboviral outbreak occurred in Italy between 2006 and 2023, by applying a consolidated Bayesian approach^{44,45}. Estimates of R_t were obtained using a Markov Chain Monte Carlo (MCMC) sampling method and applying the renewal equation to the time series of symptomatic confirmed cases by date of symptom onset. This probabilistic approach accounts for uncertainty in R_t estimates, making it particularly well-suited for real-time epidemic analysis^{8,9}. We assumed a Gamma-distributed generation time with a mean of 12.4 days and a variance of 18.5 for CHIKV¹⁴, and with a mean of 18 days and a variance of 66 for DENV⁸.”

METHODS: “Specifically, the absolute density of adult mosquitoes per hectare on a given day was modeled as a logistic function of the mean temperature observed in the preceding days, where the maximum is defined as proportional to a location-specific climatic suitability index (σ_i)²⁶. This approach assumes that seasonal variations in mosquito abundance are primarily driven by the persistence of favorable temperature conditions throughout the mosquito lifecycle and that higher abundances are expected in locations associated with greater climatic suitability, as described by the following equation for any day d and location i :

$$N_v(i, d) = \frac{\alpha \sigma_i}{(1 + e^{-k(\tilde{T}(d,i,w) - T_0)})}$$

where $\tilde{T}(d, i, w)$ represents the average temperature recorded at location i in the w days that precede d , T_0 and k are the midpoint and steepness parameters of the logistic function, and α is a scaling factor that accounts for variations in trap efficiency across mosquito capture data used for model calibration, as well as consequent rescaling factors to convert estimated captures into per-hectare abundance. The climate suitability index (σ_i) was assumed to be driven by temperature and precipitation observed in location i as in Zardini et al.²⁶ Overall, the model was informed by georeferenced presence-absence records of *Ae. albopictus* across 4,372 locations in Europe and 300 time-series of female adult mosquitoes collected between 2007 and 2018. The resulting estimates of vector abundance were combined with human density retrieved from the WorldPop data⁴⁶ to assess arbovirus transmission potential. Specifically, for each location and year of interest, we calculated the basic reproduction number (R_0) and the likelihood of onward transmission using a Susceptible, Exposed, Infectious, Recovered (SEIR) host and Susceptible, Exposed, Infectious (SEI) vector model. Both measures were computed following standard approaches as proportional to the vector-to-host ratio ($N_v(i, d)/N_h(i)$), while incorporating disease- and vector-specific parameters obtained from the literature.”

Data Availability: A dedicated data availability section is currently missing. To improve transparency and reproducibility, it is essential to provide access to input R codes, processed datasets, and relevant outputs in a public repository. This would enable other researchers to validate the findings and build upon this important work. **Reproducibility and Accessibility:** Sharing the computational tools and datasets would also align with best practices in scientific research, allowing for broader applicability and encouraging future studies to leverage this framework for risk assessment in other regions.

We apologize for the lack of information on these important points. While individual data are not publicly available due to their sensitive nature, we provide information on where aggregate data can be accessed. Regarding the sharing of code used in our analysis, a public repository with no restrictions will be made available on Zenodo or github upon publication, providing access to the code, processed data, and detailed estimates. In response to both the reviewer's comment and the journal's policy, we now provide clear statements about data and code availability as follows:

DATA AVAILABILITY

“The data contain confidential information, and public data deposition is not permitted. Due to the sensitive nature of the data, raw data can only be made available by the Istituto Superiore di Sanità (Italian National Institute of Health) through a data-sharing agreement directly with the user (contact mail: sorveglianza.arbovirosti@iss.it). Aggregated data are available at <https://www.epicentro.iss.it/arbovirosti/dashboard> and <https://www.epicentro.iss.it/arbovirosti/bollettini>.

CODE AVAILABILITY

“The statistical analysis was performed using R statistical software version 4.2.2 2022 (R Project for Statistical Computing). The code will be made available on Zenodo at [10.5281/zenodo.15100955](https://zenodo.org/records/10203374) upon publication. The analysis of reproduction numbers and the potential risk of onward transmission was performed using codes developed ad-hoc by our team in the programming language C, available at <https://zenodo.org/records/10203374>, and codes widely adopted for estimating the net reproduction number available at <https://github.com/majelli/Rt>.”

In summary, while this study is undoubtedly a valuable contribution to the field, addressing these points would further enhance its scientific rigor and accessibility, ensuring its long-term impact on the understanding and mitigation of arboviral transmission risks in Europe.

We would like to once again thank the Reviewer for their kind words of appreciation. We have addressed all their comments and believe that the manuscript has been improved in terms of clarity, transparency, and reproducibility.

Reviewer #3 (Remarks to the Author):

This article is of great interest and possibly well conducted. However, it is extremely hard to judge it because the Methods section did not explain how the analyses were performed sufficiently well. The manuscript is also challenging to read because some important information has been left out, or misplaced, throughout and many details that are not needed are in the main text and obstruct the flow. I suggest a thorough rewrite of the entire manuscript, so that I can better assess it. Following are my suggestions in each section/paragraph as I went through the manuscript.

We thank the reviewer for the time spent evaluating our manuscript and for their valuable feedback. We apologize for any lack of clarity in the description of the methods in the main text. In the original manuscript, all methodological details were described in the Supplementary Information. Following the reviewer's comment, we reorganized the manuscript to accommodate their suggestions, ensuring that methodological approaches and modeling assumptions underlying our analyses are also briefly presented in the main text, while aligning with the journal's style and guidelines.

The main changes to the text can be summarized as follows:

1. We added a new section in the *Methods* to describe how the surveillance and control of *Aedes*-borne arboviruses is carried out in Italy.
2. We added a new section in the *Results* to describe the characteristics of travel-related and autochthonous cases.
3. We included additional details in the *Methods* on how we estimate the daily absolute density of mosquitoes per hectare and the reproduction number for the two diseases.
4. We provided an explanation of the rationale behind the choice of models used for the statistical analysis of the imported cases.
5. We reorganized the text of both the *Introduction* and *Discussion* to enhance the flow of the narrative.

We decided to keep further methodological details in the Supplementary Information to maintain readability and conciseness, in accordance with the journal's requirements. Hereafter, we provide a detailed point-by-point response to all comments raised by the reviewer.

Abstract

Please expand on how exactly you analyze the data.

The words "assessed" and "leveraging" could be switched for more precise wording.

Addressed. The text now reads as follows:

"We applied generalized additive models to the records of travel-related cases to highlight the spatiotemporal patterns of disease importation, calculated reproduction numbers for six local outbreaks based on autochthonous case data and mapped

current transmission risks by applying a computational model that integrates human density, entomological, and climate data.”

Introduction

Second half of third paragraph seems off topic, starting at the fourth sentence on line 67. The first half discusses the recent outbreaks of dengue, but then the second half start a discussion about the different levels of government involved. While the first half is interesting and fits neatly in this position in the Introduction, the second is not very interesting for most readers and belongs more in the Material and Methods, perhaps in a section about description of the study area. I recommend merging the first half of the third paragraph, down to line 67 at the end of the third sentence, with the second paragraph, and either delete the second half or move it to the Material and Methods. The fourth and fifth paragraphs give little detail about the study framework or the analyses performed in this study. They both describe certain programs and systems in place at various levels of government to investigate disease outbreaks. While this may be interesting to some readers, the space in these two paragraphs would be much better used by describing the datasets used, in terms of where and when and which types of variables were collected, and by explaining the type of analyses they performed in this study and why, and their hypotheses. I would delete the fourth and fifth paragraphs, and expand the last paragraph for a much more interesting and informative introduction.

Following the reviewer’s comment, we carefully revised and reorganized the main text. Specifically, in response to the comments received from the three reviewers, we created a dedicated section in the *Methods* to describe how surveillance and control of *Aedes*-borne diseases are conducted in Italy (see the new section “*Surveillance and Control of Aedes-borne Arboviruses in Italy*”). To meet the journal’s requirements, we expanded on the data collected through human surveillance in the section “Epidemiological Data” within the *Methods*, rather than in the *Introduction*. The description of the characteristics of travel-related cases and potential outbreak index cases is now provided in the first section of the *Results* (see the new section “*Descriptive Characteristics of Travel-related Cases and Potential Outbreak Index Cases*”). Hypotheses and assumptions made in our analysis are briefly outlined for each method employed, with detailed and technical information provided in the Supplementary Information. Finally, following the Reviewer’s suggestion, we expanded the final part of the *Introduction* to anticipate the objectives, the data considered, and the methodology for the different analyses conducted, as follows:

“To address the limited epidemiological evidence quantifying arboviral epidemic risks in Europe - and given the larger CHIKV and DENV outbreaks observed in Italy compared to other European countries - we conducted a retrospective analysis of *Aedes*-borne transmission in Italy between 2006 and 2023. This includes the analysis of 1,577 travel-related and 481 autochthonous cases of CHIKV and DENV identified in the country during this period. We applied generalized additive models to the time series of travel-related cases to highlight spatiotemporal risks of disease importation and to identify temporal changes in notification delays. We estimated and compared the daily reproduction number for CHIKV and DENV, using consolidated time series of symptomatic cases for each outbreak and a model informed by mosquito capture data, incorporating season-specific temperature data to account for climatic factors.”

We thank the reviewer for the provided detailed suggestions, as we believe that this reorganization improved the readability of both the *Introduction* and the *Discussion*.

Results

Please choose a syntax form and keep it throughout for DENV/CHIKV or dengue/chikungunya.

We thank the reviewer for carefully reading the manuscript. We have revised the text and addressed this point throughout the manuscript.

No need to refer to the Method section at line 147 because this is a given.

Addressed

Describe in figure captions which software was used to generate maps, where appropriate.

As now clearly stated in the *Methods*, maps were created using QGIS software version 3.30.2, while the estimates of reproduction numbers and the potential risk of onward transmission displayed in these maps were obtained using codes developed ad-hoc by our team in the C programming language. Please note that any relevant code used to produce these estimates is already available at <https://zenodo.org/records/10203374>. Additionally, a public repository with no restrictions will be made available on Zenodo or GitHub upon publication, providing access to codes, processed data, and detailed estimates. Following the journal's requirements, this is now clarified in the *Code Availability* statement of the manuscript (rather than in the figures' captions).

I think Figure 5 could be removed or placed in supplementary material.

Addressed

From which results do you draw the conclusion at lines 162-164?

We apologize for the lack of clarity, and rephrased the sentence as follows:

“Concerning the 2007 CHIKV outbreak in Emilia Romagna, we found a decline of R_0 at the end of July 2007 as resulting from changes in temperature conditions. This may partially explain the corresponding decline of R_t found before outbreak detection (see Fig. 2).”

Figure 6 seems to contain two unrelated figures. I would put panels A and B in its own figure, and C, D, E and F in a separate one. If number of figures is restricted by the journal, deleting Figure 5 should open up space to divide Figure 6 into two.

Following the Reviewer's suggestion, we have separated Figure 6 into two distinct figures. To accommodate this change, we have moved the previous Figure 5 to the Supplementary Information.

Discussion

Discussion is currently about 3 times the length of the Introduction. Many elements of the Discussion, particularly in the few earlier paragraphs, would best be presented in the Introduction. The Discussion should mostly be where results on this study are discussed in light of past results from previous studies, except a paragraph at the beginning that serves as a reminder of the study objectives, and a paragraph at the end that serves as a summary of the study and perhaps also as a take home message. These are standard guidelines, that may not be applicable to all articles, but following them more closely here could really benefit the article as a whole and make the message flow better for the reader.

In the revised version of the manuscript, we have deleted the first four paragraphs from the *Discussion* and reorganized their content into the *Introduction*. Furthermore, the *Discussion* has been carefully revised to address each result of our analysis in the context of available evidence and previous studies, while highlighting the potential public health implications of the findings from our study. We believe that the restructuring of the main text has improved the overall readability and clarity of the manuscript.

Also, remember to always write the entire name of abbreviations at first mention, such as WHO and ISS.

Addressed

Methods

In the section starting at line 331, state the softwares and version you used for the generalized additive mixed models. Why did you use generalized additive mixed models, instead of other types of generalized mixed models? Which variables were used as random-effects variables? For example, “the interaction between these terms and the disease was also considered” is way too vague. How were the interaction terms considered? Did you include all interaction terms in the models as fixed-effects variables? Did you test correlation among them and their respective source variables?

Generalized Additive Models (GAMs) were preferred over linear models because they accommodate nonlinear effects through smooth functions of covariates. This is particularly useful for modeling travel-related cases, which are expected to exhibit seasonal patterns rather than a simple, monotonic relationship with the months. Furthermore, GAMs mitigate overfitting by employing penalized regression splines, resulting in a more robust and interpretable model. Our model assumptions can be summarized as follows:

- 1) To explore temporal trends in disease importation, we use GAMs incorporating the year and month of symptom onset, modeled as penalized cubic splines and penalized cyclic cubic splines, respectively. Interactions between these temporal variables and the disease were considered, resulting in four smoothing functions. While no random effects were considered for this analysis, an autoregressive term of order two was included to control for temporal

autocorrelation. The analysis was carried out using the function `gamm` implemented in the R package `mgcv` (R Project for Statistical Computing, software version 4.3.2).

- 2) To explore temporal changes in the notification delays, we employed a generalized additive mixed model (GAMM). Following the same approach, the year and month of symptom onset were modeled as penalized cubic splines and penalized cyclic cubic splines, respectively. The interaction between these terms and the disease was accounted for as in the GAM. In this case, the region of importation was considered as a random effect.
- 3) To investigate whether the notification delay differed between autochthonous and imported cases, we included autochthonous cases in our analysis and focused solely on regions and years where autochthonous transmission was documented. As in this case temporal trends were not the focus of our analysis, we applied Generalized Linear Mixed Models (GLMM), considering the outbreak as a random effect. The analysis was carried out using the function `glmer.nb` implemented in the R package `lme4` (R Project for Statistical Computing, software version 4.3.2).

All these points have been clarified in the text. For the Reviewer's convenience, we provide below excerpts from the revised main text addressing all the points raised,

"Generalized Additive Models (GAMs) were preferred over linear models because they accommodate nonlinear effects through smooth functions of covariates. This is particularly useful for modeling travel-related cases, which are expected to exhibit seasonal patterns rather than a simple, monotonic relationship with the months. Furthermore, GAMs mitigate overfitting by employing penalized regression splines, resulting in a more robust and interpretable model."

[...]

"Regarding quantitative covariates, the year and month of symptom onset were modeled as penalized cubic splines and penalized cyclic cubic splines, respectively. To explore the relationships between each disease and both the month and year of importation, interactions between these temporal variables and the disease were considered, resulting in four smoothing functions. Finally, an autoregressive term of order two was included in the model to control for temporal autocorrelation. The analysis was carried out in a frequentist framework using the function `gamm` implemented in the R package `mgcv` (R Project for Statistical Computing, software version 4.3.2)."

[...]

"Following a similar approach, we investigated possible temporal changes in the delay associated with the notification of imported cases to the central health authority, defined as the time between the symptom onset of cases and their reporting to the National Surveillance System through the ISS surveillance platform. In this case, we employed a generalized additive mixed model (GAMM), by assuming that the notification delay follows a Negative Binomial distribution with log link. Whether the case was dengue or chikungunya was included as a qualitative covariate. The year and month of symptom onset were modeled as penalized cubic splines and penalized cyclic cubic splines, respectively. The interaction between these terms and the disease was accounted for as in the GAM. The region of importation was considered as a random effect."

[...]

“We finally investigated whether the notification delay differed between autochthonous and imported cases by means of Generalized Linear Mixed Models (GLMM), assuming a Negative Binomial distribution with log link. To do this, we included autochthonous cases in our analysis and focused solely on regions and years where autochthonous transmission was documented (chikungunya: 2007 in Emilia-Romagna, 2017 in Lazio and Calabria; dengue: 2020 in Veneto, 2023 in Lazio and Lombardy). Two categories of autochthonous cases were considered: those with symptom onset preceding the outbreak detection and those occurring afterward. Considered covariates included the disease and case classification (i.e., imported, autochthonous preceding the outbreak detection, autochthonous following the outbreak detection); the outbreak was considered as a random effect. The analysis was carried out in a frequentist framework using the function `glmer.nb` implemented in the R package `lme4` (R Project for Statistical Computing, software version 4.3.2).”

To address potential multicollinearity, we measured concurvity among model components using the `concurvity` function in the `mgcv` package (R Project for Statistical Computing, software version 4.3.2). Concurvity occurs when a smooth term in the model can be approximated by one or more of the other smooth terms within the same model and can be interpreted as a generalization of collinearity. Although there is no clear statistical consensus, most authors suggest that concurvity values below 0.5 or 0.8 are considered acceptable. In our case, all concurvity values were below 0.4, except for one, which reached 0.606 (see new Tables S2 and S5). Visual inspection of the smoothing shapes and confidence intervals, along with investigation of pairwise concurvity (with all concurvity values resulting <0.4), did not indicate the need to drop any interaction terms due to multicollinearity. We have now included the concurvity results in the Supplementary Information (Tables S2 and S5).

At the paragraph starting at line 351, you need to add more details on how you ran this analysis. How did you investigate the comparison between autochthonous and imported cases? Which software, which software version, what analysis type, and if modelling, which type of model did you use? These are more important than most of what is currently written in this paragraph. If space is limited, most of the current paragraph could be relegated to the Appendix, as already referred to in the same paragraph.

We apologize for the lack of details on this point. In the original manuscript, all methodological details, including the formulas for each statistical model used in our study, were provided in the Supplementary Information. Following the Reviewer's suggestion, we have revised and reorganized the text to clarify our modeling assumptions and briefly describe the methodological approaches in the main text while ensuring alignment with the journal's style and guidelines. As now explicitly stated in the main text (see the response to the previous comment), we explored potential differences in notification delay between autochthonous and imported cases using a Generalized Linear Mixed Model (GLMM) with a Negative Binomial distribution and a log link. The considered covariates were disease type (dengue vs. chikungunya) and case classification (imported, autochthonous before outbreak detection, autochthonous after outbreak detection). The outbreak was included as a random effect.

At the paragraph starting at line 360, describe the consolidated Bayesian approach in a more precise way, and explain why you use this approach? Also, state the software, version and functions you used. The section is small and should have more details in the main text and less in the Appendix.

We have now expanded the description of the method used to estimate the net reproduction number as follows:

“Estimates of R_t were obtained using a Markov Chain Monte Carlo (MCMC) sampling method and applying the renewal equation to the time series of symptomatic confirmed cases by date of symptom onset. This probabilistic approach accounts for uncertainty in R_t estimates, making it particularly well-suited for real-time epidemic analysis^{8,9}. We assumed a Gamma-distributed generation time with a mean of 12.4 days and a variance of 18.5 for CHIKV¹⁴, and with a mean of 18 days and a variance of 66 for DENV⁸. The analysis was performed using codes widely adopted for estimating the net reproduction number available at <https://github.com/majelli/Rt>. More details are provided in the Supplementary Information.”

In the prospective analysis starting at line 367, very few details were given by the authors, which makes it very difficult to assess how the analysis was conducted and the extent of its limitations. While I cannot fully assess how this analysis was done and therefore if relevant, a large difference in resolution among the various explanatory variables used may be a source of spatial biases. Also, data use to estimate *Aedes* spp. abundance was not described. Did you use mosquito capture and arbovirus testing data, and of which species exactly? If this analysis relies only on human cases, describe how the estimates are still accurate and meaningful, and otherwise I believe this limitation should be noted in the Discussion.

In response to the Reviewer’s concerns, we expanded the *Methods* section to provide more technical details on how model estimates of R_0 from entomological and climate data were obtained, including the equation used to determine the daily absolute density of mosquitoes per hectare and a brief description of the entomological data used for model calibration. Additionally, we revised the text throughout the manuscript (e.g., changing the titles of different sections, revising the abstract and the final part of the *Introduction*) to emphasize that this model relies solely on human density—rather than human case records—combined with entomological and climate data. As in the previous version of our manuscript, further details are provided in the Supplementary Information.

The revised main text now reads as follows:

“Specifically, the absolute density of adult mosquitoes per hectare on a given day was modeled as a logistic function of the mean temperature observed in the preceding days, where the maximum is defined as proportional to a location-specific climatic suitability index (σ_i)²⁶. This approach assumes that seasonal variations in mosquito abundance are primarily driven by the persistence of favorable temperature conditions throughout the mosquito lifecycle and that higher abundances are expected in locations associated with greater climatic suitability, as described by the following equation for any day d and location i :

$$N_v(i, d) = \frac{\alpha \sigma_i}{(1 + e^{-k(\tilde{T}(d, i, w) - T_0)})}$$

where $\tilde{T}(d, i, w)$ represents the average temperature recorded at location i in the w days that precede d , T_0 and k are the midpoint and steepness parameters of the logistic function, and α is a scaling factor that accounts for variations in trap efficiency across mosquito capture data used for model calibration, as well as consequent rescaling factors to convert estimated captures into per-hectare abundance. The climate suitability index (σ_i) was assumed to be driven by temperature and precipitation observed in location i as in Zardini et al.²⁶ Overall, the model was informed by georeferenced presence-absence records of *Ae. albopictus* across 4,372 locations in Europe and 300 time-series of female adult mosquitoes collected between 2007 and 2018. The resulting estimates of vector abundance were combined with human density retrieved from the WorldPop data⁴⁶ to assess arbovirus transmission potential. Specifically, for each location and year of interest, we calculated the basic reproduction number (R_0) and the likelihood of onward transmission using a Susceptible, Exposed, Infectious, Recovered (SEIR) host and Susceptible, Exposed, Infectious (SEI) vector model. Both measures were computed following standard approaches as proportional to the vector-to-host ratio ($N_v(i, d)/N_h(i)$), while incorporating disease- and vector-specific parameters obtained from the literature. We compared estimates of R_0 resulting by assuming *Ae. albopictus* as the only vector for arboviral transmission in Italy with R_t values estimated from human cases identified during the different outbreaks.”

Finally, please note that we added a statement in the *Discussion* to acknowledge that different spatial resolutions could be a potential source of bias for our model estimates:

“Although temperature and precipitations are expected to be relatively homogeneous within a single municipality, differences in the spatial resolution of temperature, precipitation, and human density data might lead to potentially biased estimates of the transmission risk. This effect could be especially relevant in areas with strong climatic heterogeneity, such as mountain areas, where, however, the risk of arboviral transmission is consistently low.”

Daily temperature records at a resolution of 0.1 x 0.1 degrees is very coarse. You may obtain land surface temperature raster files at a 1km x 1km resolution from USGS. Did you also consider precipitation, which are very important to predict mosquito habitat?

We apologize for the lack of clarity. The model used to estimate mosquito abundance is an extension of a recently published computational model (Zardini et al., *The Lancet Planetary Health*, 2024), which assumes that mosquito relative density is proportional to climate suitability for *Aedes albopictus*. This suitability was estimated using temperature and precipitation data retrieved from WorldClimate at a spatial resolution of 1 km x 1 km. Since these data represent long-term environmental conditions (averages for the years 1970–2000), rather than intra- and inter-annual temperature variations, in this work, we have incorporated E-OBS daily mean temperature data to account for seasonal and inter-annual variations in

transmission risks. To better clarify the role of climate suitability, we have now reported the equation for the modeling function that estimates the daily absolute density of mosquitoes per hectare for any specific location (i), emphasizing that σ_i depends on precipitation and climate data. All of this information is further detailed in the Supplementary Information.

Following the Reviewer suggestion, we considered the possibility of using raster files from USGS. Consequently, we extracted daily raster files for Italy at a spatial resolution of 1 km x 1km from Terra Moderate Resolution Imaging Spectroradiometer (MODIS) Land Surface Temperature/Emissivity Daily (MOD11A1) Version 6.1 (<https://lpdaac.usgs.gov/products/mod11a1v061/>) and Aqua MODIS Land Surface Temperature/Emissivity Daily (MYD11A1) Version 6.1 (<https://lpdaac.usgs.gov/products/myd11a1v061/>). We conducted an exploratory analysis on the daily land surface temperature (LST) records for both daytime and nighttime, collected by MODIS Terra and Aqua over Italy from June 1, 2023, to November 1, 2023 (154 days), and identified the following limitations:

1. Most patches were associated with missing data during either daytime or nighttime, possibly due to cloud cover. Considering LST data from the Terra satellite, 99.5% of the patches had more than 30 days of missing daytime LST data (approximately 1/5 of the analyzed time window) and 60.1% had more than 50 days missing (around 1/3 of the analyzed time window). Considering LST nighttime data from the Terra satellite, data was missing for more than 30 nights in 99.9% of the patches, and for more than 50 nights in 82.5% of the patches. Similarly, for the Aqua satellite, daytime LST data was missing for more than 30 days in 99.9% patches and for more than 50 days in 79.9%. For nighttime Aqua data, 99.9% and 65.9% of patches had missing data for more than 30 and 50 nights, respectively. We included four maps (shown below) to display the number of days and nights with missing LST data for each patch, considering either the Terra or Aqua satellite.

Terra daytime

Terra nighttime

Aqua daytime

Aqua nighttime

- Several patches had missing data for consecutive days within the selected time window. Daytime LST data from the Terra satellite were missing for more than 7 consecutive days in 46.3% of the patches, while the corresponding percentage for Terra nighttime LST data was 60.6%. For the Aqua satellite, 65.9% of the patches had more than 7 consecutive days with missing daytime LST data, and for Aqua nighttime LST data, the percentage was 42.8%. The four maps displayed below show the maximum number of consecutive days with missing data for each patch, considering either the Terra or Aqua satellite.

Terra daytime

Terra nighttime

Aqua daytime

Aqua nighttime

3. Terra LST data was missing for more than 30 days in both daytime and nighttime in 70.4% of patches; the corresponding proportions for the Aqua satellite was 72.0%. Patches where LST data from both satellites were missing for the same daytime period for more than 30 days accounted for 83.1% of the total. For the nighttime, this percentage was 68.1%.
4. This situation is even more pronounced when focusing on the specific areas where the outbreaks occurred. For instance, Terra daytime LST data were missing for all patches in the municipality of Castiglione d'Adda for 57 days (70 days for nighttime data), while Aqua daytime and nighttime data were missing for 46 and 57 days, respectively. In Montecchio Maggiore, Terra daytime LST data were missing for all patches in the municipality for 62 days, and Terra nighttime LST data were missing for 87 days. For Aqua data, daytime and nighttime LST were missing for 68 and 48 days, respectively.

In sum, most patches had missing LST data from both satellites for large portions of the considered time window, possibly for more than 7 consecutive days. This issue occurs for both daytime and nighttime LST data, and it happens that patches lack data from both satellites on the same day. Consequently, using this data would require several assumptions to fill in the missing values. The E-OBS daily mean temperature data we used, despite being available at a coarser spatial resolution, are instead complete for each patch on each day of the outbreak years.

In addition, E-OBS data are derived from the interpolation of station-based temperature records, which may provide more accurate representations of relevant temperature conditions for the purposes of our work than LST data. Specifically, we compared E-OBS daily mean temperature data and the Terra and Aqua LST data with the mean daily temperature recorded in 2023 by a weather station near to the municipality of Castiglione d'Adda, where a DENV outbreak occurred in 2023 (coordinates of the station: N 45.233577°; E 09.665346°). To estimate the daily

mean temperature from LST data, we averaged the daytime and nighttime LST values when both were available. Figures shown below display:

1) the daily mean temperature recorded by the weather station compared with the daily mean temperature reported by either E-OBS, Terra, or Aqua satellite averaged patches constituting the municipality (point: mean; bar: 2.5–97.5 percentile range). We computed the average on the municipality only for the days where all patches within the municipality had no missing values;

2) The absolute error of E-OBS, Terra, and Aqua daily mean temperatures, averaged across the municipality, was calculated with respect to the daily mean temperatures recorded by the considered weather station. The error was computed as the sum of the absolute differences between the E-OBS/Terra/Aqua data and the weather station data, considering days when data from E-OBS and Terra, E-OBS and Aqua, and all three (E-OBS, Terra, and Aqua) were available. We displayed the mean error (point) along with 2.5–97.5 percentile range (line).

This comparison not only emphasizes the high frequency of missing LST data but also highlights that using daily mean temperature estimates derived from Terra or Aqua instead of E-OBS would not improve the alignment with weather station data.

For all these reasons – considering also that climate suitability is estimated using precipitation and temperature data at a spatial resolution of 1 km x 1 km - we decided to keep using E-OBS daily data instead of MODIS LST data, despite its finer resolution.

Nonetheless, we appreciate the Reviewer’s suggestion, as it aimed to enhance the precision of our model estimates and provided an opportunity to explore a worthwhile possibility.

Global Human Settlement Layer is more a land use dataset, showing built-up density, which is somewhat correlated with human population density. If the authors need actual human population density, Worldpop might be more appropriate.

We thank the Reviewer for the valuable comment. In the revised version of the manuscript, we re-ran the model using human density data for Italy from the WorldPop database at a spatial resolution of 100 m x 100 m, obtaining estimates of the basic reproduction number and the likelihood of onward transmission at this higher resolution. The main conclusions of the study remained largely unchanged from the original version. However, we once again appreciate the Reviewer’s suggestion, as it has helped to enhance the precision of our model estimates.

The paragraph starting at line 391 is very important and useful, but I believe these details should be expanded on and added throughout the Methods section.

We have now expanded the details on the codes and software used for each analysis, specifying the functions and software utilized throughout the *Methods* section and in the Supplementary Information. Additionally, we will make a public repository available on Zenodo or GitHub upon publication to provide access to the code, processed data, and detailed estimates. Please note that, following the journal's guidelines, a new section on data and code availability has also been included in the main text.